systems biology

glioblastoma, drug resistance, O-6-methylguanine-DNA methyltransferase, stochastic modelling, multi-scale model, phenotypic selection

**Author for correspondence:**
Ayoub Lasri
e-mail: lasriay@gmail.com

# Phenotypic selection through cell death: stochastic modelling of O-6-methylguanine-DNA methyltransferase dynamics

Ayoub Lasri[1], Viktorija Juric[1], Maité Verreault[2], Franck Bielle[3], Ahmed Idbaih[3], Alexander Kel[4,5], Brona Murphy[1] and Marc Sturrock[1]

[1]Department of Physiology and Medical Physics, Royal College of Surgeons in Ireland, York House, Dublin, Ireland
[2]Inserm U 1127, CNRS UMR 7225, Sorbonne Université, Institut du Cerveau et de la Moelle épinière, ICM, 75013 Paris, France
[3]Sorbonne Université, Inserm, CNRS, UMR S 1127, Institut du Cerveau et de la Moelle épinière, ICM, AP-HP, Hôpitaux Universitaires La Pitié Salpêtrière – Charles Foix, Service de Neurologie 2-Mazarin, 75013 Paris, France
[4]Department of Research and Development, geneXplain GmbH, Wolfenbüttel 38302, Germany
[5]Laboratory of Pharmacogenomics, Institute of Chemical Biology and Fundamental Medicine, Novosibirsk 630090, Russia

 AL, 0000-0003-0806-9072; VJ, 0000-0003-2916-9080;
FB, 0000-0001-6564-6388; AI, 0000-0001-5290-1204;
BM, 0000-0002-6740-8858

Glioblastoma (GBM) is the most aggressive malignant primary brain tumour with a median overall survival of 15 months. To treat GBM, patients currently undergo a surgical resection followed by exposure to radiotherapy and concurrent and adjuvant temozolomide (TMZ) chemotherapy. However, this protocol often leads to treatment failure, with drug resistance being the main reason behind this. To date, many studies highlight the role of O-6-methylguanine-DNA methyltransferase (MGMT) in conferring drug resistance. The mechanism through which MGMT confers resistance is not well studied—particularly in terms of computational models. With only a few reasonable biological assumptions, we were able to show that even a minimal model of MGMT expression could robustly explain TMZ-mediated drug resistance. In particular, we showed that for a wide range of parameter values constrained by novel cell growth and viability assays, a model accounting for only stochastic gene expression of MGMT coupled with cell growth, division, partitioning and death was able to exhibit phenotypic

selection of GBM cells expressing MGMT in response to TMZ. Furthermore, we found this selection allowed the cells to pass their acquired phenotypic resistance onto daughter cells in a stable manner (as long as TMZ is provided). This suggests that stochastic gene expression alone is enough to explain the development of chemotherapeutic resistance.

# 1. Background

Glioblastoma (GBM) is the most aggressive malignant primary brain tumour with a median overall survival of 15 months. Initial treatment for newly diagnosed GBM patients consists of maximal surgical resection with the aim to achieve a gross-total resection (GTR)—all of the visible tumour is removed and no apparent tumour is observed post-surgery using MRI with contrast. Following surgery, adjuvant treatment is applied to the remaining cancer cells and the so-called Stupp protocol is widely used nowadays [1]. According to this protocol, patients receive concomitant radiotherapy (RT) and temozolomide (TMZ) followed by adjuvant TMZ. RT consists of fractionated focal irradiation at a dose of 2 Gy per fraction given once daily, 5 days per week over a period of six weeks, for a total dose of 60 Gy. It is delivered to the gross tumour volume with a 2 to 3 cm margin for the clinical target volume. Concomitant chemotherapy consists of TMZ at a dose of 75 mg m$^{-2}$ per day, given 7 days per week during RT. Upon completion of chemo- and RT, there should be a four-week break. After that, patients should receive up to six cycles of adjuvant TMZ, given for 5 days every 28 days, with a starting dose of 150 mg m$^{-2}$ for the first cycle. This dose is subsequently increased to 200 mg m$^{-2}$ at the beginning of the second cycle, as long as there are no haematologic toxic effects.

Despite the above-mentioned standard therapy, which is considered to be the 'gold standard' for GBM patients to date, patient prognosis remains dismal. Several factors contribute to this poor survival, including but not limited to: the patient's condition, the tumour location as well as genetic instability within GBM cells. The anatomical location of the tumour quite often makes GBM very difficult to treat. Another potential mechanism of treatment failure is the blood-brain barrier, which consists of astrocytes and specialized endothelial cells with tight junctions that serves to restrict brain uptake of drugs [2]. One of the most critical cellular markers of GBM, obtained post-diagnosis upon pathological analysis, is the methylation status of O-6-methylguanine-DNA methyltransferase (MGMT), which is under epigenetic control of its expression through its corresponding promoter [3–6].

MGMT is a DNA repair protein that is involved in the cellular defence against toxicity from alkylating agents [7,8]. Some clinical studies have linked a positive methylation status of the promoter of MGMT with greater sensitivity to TMZ [3,9]. The relevance of MGMT methylation as a biomarker has been strengthened by the widely accepted application of the consensus reached in Phase III clinical trials jointly conducted by European and North American research networks that were summarized in the Stupp protocol [10]. Kitange *et al.* [11] associated the transition from methylated to unmethylated status to TMZ activity and demonstrated an inverse relationship between MGMT protein levels and TMZ sensitivity, with high levels of MGMT expression invariably associated with TMZ resistance. In [12], Kitange *et al.* evaluated the relationship between MGMT protein expression and tumour response to TMZ and have shown that MGMT protein expression increases in response to TMZ. In other studies [13,14], low MGMT mRNA expression has been found to be predictive of a better response to TMZ, consistent with the elevated methylation pattern of MGMT promoter. However, in [15], it was shown that there is a discordance between methylation status and expression level of MGMT. In particular, it was shown that a hypermethylated MGMT promoter could coexist with high MGMT expression. Therefore, the mechanism by which MGMT confers resistance is not well understood and could benefit from the development of computational models.

The control of transcription is mediated by factors that bind at upstream promoter elements or influence the binding of other molecules to *cis*-regulatory elements within or near the promoter. Because such binding events are the result of random encounters between molecules, some of which are present in small numbers, the biochemical processes that regulate transcription initiation are inherently stochastic [16]. Stochasticity in gene expression is linked to variability in cellular characteristics, i.e. identical cells exposed to the same environmental conditions can show significant variation in molecular content and differences in phenotypic characteristics [17–19]. In order to model this variability, stochastic modelling approaches help link single-cell characteristics to cell population behaviour, and in our study, we use this to link MGMT gene expression to GBM resistance. Multi-scale models provide access to systems behaviours that are not observable using single-scale techniques [20]. For example, one can link noise in gene expression, cell division and partitioning to the appearance of certain phenotypes which confer resistance [21,22].

Phenotypic differences can be directly linked to the noisy molecular nature of regulatory circuits [23–25]. Unlike genetic variations that can lead to irreversible mutations, these different sources of phenotypic variability are not transmitted to the daughter cells in an integral manner allowing populations to recover from environmental stress on faster timescales than traditional genetic changes. As such, they allow cells to try out different survival strategies faster and can reverse back to a pre-stress state more rapidly. In [26], Mora et al. showed how in a constant-sized population, an mRNA species which dictates the growth rate of the cell can be selected for and lead to its distribution being upregulated. Phenotypic selection may be responsible for the rise of a resistant cell population. In [27], an analysis was presented of the rapid 'appearance' of the multi-drug resistance protein (MDR) phenotype and of multi-drug resistance protein 1 (MDR1) expression following chemotherapy. This provided evidence that this early drug resistance phenotype can be induced by a non-Darwinian selection, independent of genetic selection. In a more recent study [28], El Meouche et al. studied the multiple antibiotic resistance activator (MarA) in bacteria and linked the stochastic gene expression to the appearance of a multi-drug resistance phenotype. They found that MarA overexpression increases antibiotic resistance in population measurements and that MarA variability is correlated with survival in the presence of carbenicillin within an isogenic E. coli population. Finally, Ciechonska et al. [29] recently showed that emergent gene expression or phenotypic selection requires global positive feedbacks between cell growth and gene expression as well as revealing that an antibiotic resistance gene displays a linear dose-responsive upregulation in proportion to antibiotic concentration.

In this work, we seek to capture all these existing biological findings in a single model which will allow us to probe the relationship between MGMT expression, TMZ administration and the viability of the cell population. By formulating this relationship in a mathematical model, we are able to piece together different experimental findings and probe the system systematically. We begin by explaining our general model of MGMT dynamics, describing our modelling assumptions regarding the intracellular processes, cell growth, division, partitioning and cell death. We explore the dynamical behaviour of this model and in particular find parameter sets which can exhibit drug resistance. We then study these parameter sets in detail and establish parameter relationships and model characteristics associated with drug resistance. Next, we present experimental results obtained using a patient-derived GBM cell line which allows us to estimate the cell growth rate and the impact of TMZ on cell viability. Finally, we calibrate and validate our model using this experimental data.

# 2. Methods

## 2.1. Cell culture

N15-0385 patient-derived GBM cell line was established by the GlioTex team (GBM and Experimental Therapeutics) in the Institut du Cerveau et de la Moelle epiniere (ICM) laboratory. Cells were cultured in DMEM-F12 medium (Gibco Life Technologies) containing B27 supplement 50× (2%, Gibco Life Technologies), human bFGF (20 ng ml$^{-1}$, Peprotech), human EGF (20 ng ml$^{-1}$, Peprotech), penicillin (100 U ml$^{-1}$, Sigma-Aldrich), streptomycin (100 mg ml$^{-1}$, Sigma-Aldrich), heparin (5 µg ml$^{-1}$, Alfa Aesar) and maintained in a humidified incubator at 37°C and 5% $CO_2$. Cells were routinely tested for mycoplasma infection. For cell growth curve, cells were dissociated using Accutase (Thermo Fisher Scientific), and seeded into six-well cell culture plates ($8 \times 10^4$ cells/well). Cells were counted each day using a Countess automated cell counter (Thermo Fisher Scientific, Waltham, MA, USA) and growth curves determined from live cell numbers over a 144 h period.

## 2.2. Cell viability assay

N15-0385 cells were plated in 96-well plates (3000 cells/well) and treated with increasing concentration of TMZ (0–800 µM; Sigma-Aldrich) for 72, 96, 120 and 144 h. Following treatment, WST-1 reagent (Sigma-Aldrich) was added in 1 : 10 final dilution, according to the manufacturer's instructions. WST-1 salt is cleaved to a soluble formazan dye by a NAD(P)H-dependent reaction in viable cells. Plates were incubated for 3 h in a humidified incubator at 37°C and 5% $CO_2$ and the absorbance of each sample was measured at 450 and 620 nm using a microplate reader (GENios, Tecan, Weymouth). The absorbance was proportional to the number of viable cells and expressed relative to control-treated samples.

## 2.3. Modelling cell growth and division

In order to estimate the growth rate associated with our growth data (figure 4a), we fitted both an exponential and logistic growth curve. We found the exponential growth curve to be the more probable model of the data after using Bayesian model comparison. Hence, we modelled GBM cell growth using the following exponential growth equation:

$$\frac{dV_i(t)}{dt} = \mu_i(t)V_i(t), \tag{2.1}$$

where $V_i(t)$ is the size of cell $i$ at time $t$ and $\mu_i(t)$ is the growth rate of cell $i$ at time $t$. Prior to TMZ administration, we considered a static environment that enabled constant growth, which can be modelled as follows

$$\mu_i(t) = \mu_0, \quad \text{when } t < t_{in}, \tag{2.2}$$

where $t_{in}$ indicates when the TMZ administration begins.

In the presence of TMZ, Hoa et al. [30] reported that TMZ inhibits cell growth, hence, we assumed that TMZ affects cell growth in a detrimental manner as per the following equation:

$$\mu_i(t) := f_1 = \frac{\mu_0}{1 + [TMZ]/I_{50}}, \quad \text{when } t \geq t_{in}, \tag{2.3}$$

where $I_{50}$ stands for the half-inhibition concentration and $[TMZ]$ for TMZ concentration inside the cell at time $t$.

The analytical solution of the constant growth case can be written as

$$V_i(t) = V_i(0)\exp(\mu_0 t), \tag{2.4}$$

and this is the expression we used to update the cell volume dynamically. Cells were assumed to grow from approximately $V = 15 \times 10^{-13}$ l to approximately $V = 30 \times 10^{-13}$ l [31,32] and cell partitioning was modelled as a binomial process [33]. Mammalian cells are known to undergo division in a noisy, asymmetric manner [34]. To capture this variability, we adopt the approach of [35,36] where the final volume of the cell at generation $n$ was found to follow a noisy linear map, i.e. the final volume $V_F$ of a given cell was assumed to follow

$$V_F = aV_I + b + \eta_1, \tag{2.5}$$

where $V_I$ is the initial volume of the cell, $a$ and $b$ are linear function parameters; we note that $a$ and $b$ have the same value for all cells, and $\eta_1$ is the final volume noise. We sampled $\eta_1$ from $\mathcal{N}(0, \sigma_1)$. Recent papers have shown that using Gaussian noises in mathematical oncology can lead to biologically paradoxical results [37,38]. Though we did not encounter such issues, we also used a bounded noise approach for our general and data-constrained model simulations, where if a final volume generated was outside some finite range ($29.25 \times 10^{-13}$ l, $30.75 \times 10^{-13}$ l), the distribution was resampled. The dividing cell of volume $V_F$ gives rise to two daughter cells with volumes $V_{I_1}$ and $V_{I_2}$ defined by

$$V_{I_1} = V_F \times \eta_2 \tag{2.6}$$

and

$$V_{I_2} = V_F \times (1 - \eta_2), \tag{2.7}$$

where $\eta_2$ represents division noise and is sampled from $\mathcal{N}(0.5, \sigma_2)$. We assumed the contents of the cell are binomially distributed between daughter cells upon division [39]. In daughter cell 1, the contents (mRNA molecules, protein molecules) are inherited by sampling from a binomial distribution with probability $p = \eta_2$ while daughter cell 2 inherits the remainder of the contents. This can allow for a larger chance of daughter cell 1 to inherit more mRNA and protein molecules if $\eta_2 > 0.5$. We note that $\eta_1$ and $\eta_2$ embed both intracellular stochastic phenomena and also the stochastic influence of extracellular signals. As in [26], in order to keep the population size constant, after a cell division event the new offspring displaces another cell in the population picked at random. In the event of cell death, we simply removed that cell from the population. We note that following cell division in a population with some dead cells, we also tried preferentially replacing dead cells. Adopting this strategy yielded similar results to using random replacement (where both dead cells or living cells could be replaced). Simulating a constant-sized population is computationally cheaper than simulating

a growing population and leads to more accurate results than using an isolated lineage-based approach. In [40], it was shown that modelling the cells as isolated lineages can significantly overestimate the mean number of molecules and underestimate intrinsic noise (in the absence of any cell death). A similar approach of tracking a constant population of cells was also used in Bertaux *et al.* [36] and Ciechonska *et al.* [29]. DNA and DNA replication are not explicitly modelled and we assume that upon division daughter cells each inherit one copy of the genes (i.e. there are no mutations introduced). For all our simulations, we simulated a maximum number of K cells (we set $K = 10\,000$). The fraction of viable cells at time $t$ was computed as follows: $N(t) = M(t)/K$ where $M(t)$ is the number of living cells. We defined the cell viability recovery as the difference of the minimal number of living cells following TMZ addition and the final number of cells at the end of the simulation, i.e. $N(t_f) - \min(N(t))$ where $t_f$ is the final time point and min() function returns the minimum value of $N(t)$ across time.

## 2.4. Modelling intracellular processes

### 2.4.1. Full mass action reaction model of MGMT dynamics

We specify here the full mass action model. This model includes an explicit DNA damage state which we denote as 'DNA$_{\mathrm{DAM}}$'. We assumed that MGMT mRNA is constitutively expressed at a rate $b_1$ and is translated into protein at a rate $b_2$. Since MGMT is known to be stable [41], we did not explicitly account for MGMT degradation. Nevertheless, we investigated the impact of accounting for protein degradation and found that our results did not change significantly by including protein degradation (see electronic supplementary material, figure S1). We assume that MGMT mRNA degrades at a rate $d$. We made exactly the same assumptions for the reference protein (REF) which acts as a control. This control allowed us to rule out any global upregulation processes whereby every gene in the cell is upregulated; i.e. we are interested in a selective phenotypic selection, where only the gene which confers protection to the cell is upregulated. This is consistent with results obtained in bacteria studies which used a similar control for their study of HisC dynamics in a histidine depleted environment [42]. We assumed that DNA is damaged by TMZ with rate $k_d$ and that MGMT can repair damaged DNA with rate parameter $s$. The reactions of the model are as follows:

$$\mathrm{DNA}_i \xrightarrow{k_d[\mathrm{TMZ}]} \mathrm{DNA}_{\mathrm{DAM}_i}, \tag{2.8}$$

$$\mathrm{DNA}_{\mathrm{DAM}_i} \xrightarrow{s\mathrm{MGMT}_i} \mathrm{DNA}_i, \tag{2.9}$$

$$\mathrm{DNA}_i \xrightarrow{b_1} \mathrm{mRNA\_MGMT}_i + \mathrm{DNA}_i, \tag{2.10}$$

$$\mathrm{DNA}_{\mathrm{DAM}_i} \xrightarrow{b_1} \mathrm{mRNA\_MGMT}_i + \mathrm{DNA}_{\mathrm{DAM}_i}, \tag{2.11}$$

$$\mathrm{mRNA\_MGMT}_i \xrightarrow{d} \emptyset, \tag{2.12}$$

$$\mathrm{mRNA\_MGMT}_i \xrightarrow{b_2} \mathrm{MGMT}_i + \mathrm{mRNA\_MGMT}_i, \tag{2.13}$$

$$\mathrm{DNA}_i \xrightarrow{b_1} \mathrm{mRNA\_REF}_i + \mathrm{DNA}_i, \tag{2.14}$$

$$\mathrm{DNA}_{\mathrm{DAM}_i} \xrightarrow{b_1} \mathrm{mRNA\_REF}_i + \mathrm{DNA}_{\mathrm{DAM}_i}, \tag{2.15}$$

$$\mathrm{DNA}_{\mathrm{DAM}_i} \xrightarrow{k_{d2}} \text{cell death}, \tag{2.16}$$

$$\mathrm{mRNA\_REF}_i \xrightarrow{d} \emptyset, \tag{2.17}$$

$$\mathrm{mRNA\_REF}_i \xrightarrow{b_2} \mathrm{REF}_i + \mathrm{mRNA\_REF}_i, \tag{2.18}$$

$$\mathrm{MGMT}_i \xrightarrow{d_2} \emptyset \tag{2.19}$$

and

$$\mathrm{REF}_i \xrightarrow{d_2} \emptyset, \tag{2.20}$$

where a subscript $i$ denotes the cell number (see electronic supplementary material for more details about reactions (2.19) and (2.20)). By making the approximation that the DNA alkylation dynamics [43] occur on a much faster timescale than the kinetics of protein and mRNA species, we made a pseudo-steady-state approximation about the level of DNA$_{\mathrm{DAM}}$. This is justified in electronic supplementary material, figure S7(B), as we found that the full model deviated only from the pseudo-steady-state model as the DNA repair rate became very small. This allowed us to compute a simplified cell death rate.

We assumed there is only one copy of the MGMT DNA per cell and that the following conservation equation holds:

$$\text{DNA}_i = 1 - \text{DNA}_{\text{DAM}_i}. \tag{2.21}$$

The corresponding mean-field ordinary differential equation (ODE) derived from applying the law of mass action to reactions (2.8) and (2.9) for $\text{DNA}_{\text{DAM}}$ in cell $i$ can be written as

$$\frac{d\text{DNA}_{\text{DAM}_i}}{dt} = k_d \cdot [\text{TMZ}] \cdot \text{DNA}_i - s \cdot \text{MGMT}_i \cdot \text{DNA}_{\text{DAM}_i}. \tag{2.22}$$

At pseudo-steady state, the left-hand side becomes 0, yielding

$$0 = k_d \cdot [\text{TMZ}] \cdot \text{DNA}_i - s \cdot \text{MGMT}_i \cdot \text{DNA}^*_{\text{DAM}_i}, \tag{2.23}$$

where $\text{DNA}^*_{\text{DAM}_i}$ stands for $\text{DNA}_{\text{DAM}_i}$ at the pseudo-steady state. Rearranging this equation for the pseudo-steady-state value of $\text{DNA}_{\text{DAM}}$, we find

$$s \cdot \text{MGMT}_i \cdot \text{DNA}^*_{\text{DAM}_i} = k_d \cdot [\text{TMZ}] \cdot \text{DNA}_i. \tag{2.24}$$

Applying the conservation equation (2.21), we obtain

$$\text{DNA}^*_{\text{DAM}_i} = \frac{k_d \cdot [\text{TMZ}]}{k_d \cdot [\text{TMZ}] + s \cdot \text{MGMT}_i}. \tag{2.25}$$

Combining this term with the reaction defined in (2.16) gives us an expression for the cell death rate in terms of TMZ concentration and MGMT levels, i.e.

$$f_2 = k_{d2} \frac{k_d \cdot [\text{TMZ}]}{k_d \cdot [\text{TMZ}] + s \cdot \text{MGMT}_i}. \tag{2.26}$$

The cell death reaction in the reduced model is given by

$$\emptyset \xrightarrow{\epsilon + f_2} \text{cell death}, \tag{2.27}$$

where $\epsilon$ is a basal cell death rate and the cell death reaction in the general model is defined by

$$\emptyset \xrightarrow{f_2} \text{cell death}. \tag{2.28}$$

We set $k_{d2} = 1 \, \text{h}^{-1}$ for simplicity. Using these assumptions and simplifications, the reduced model we simulated is defined by chemical reactions (2.10), (2.12)–(2.14), (2.17), (2.18) and (2.27). We refer to the case where no data calibration is considered as the general model and use (2.28) for the cell death reaction in this case. Both models are illustrated in figure 1.

### 2.4.2. Stochastic gene expression

Gene expression is a stochastic process, with randomness in transcription and translation leading to significant cell-to-cell variations in mRNA and protein levels [44]. Monte Carlo simulations for well-mixed chemically reacting systems, also known as the stochastic simulation algorithm (SSA) [45] is a commonly used computational tool in systems biology which was first introduced in [46]. The SSA is particularly applicable when modelling populations with low copy numbers and can become overly computational expensive when simulating systems with large numbers of molecules [47]. To overcome these limitations an approximate accelerated stochastic algorithm was introduced in [48], also known as Tau-leaping or the Poisson-distribution-$\tau$ leap algorithm. In order to simulate the intracellular reactions defined by reactions (2.10), (2.12)–(2.14), (2.17), (2.18) and (2.27), we adopted a simple approach. Between fixed time steps (6 min), the cell volume was considered constant, and the Gillespie algorithm (Algorithm 1) was used to simulate the stochastic molecular reactions. The cell volume was computed according to the instantaneous exponential growth rate, and we checked whether cell division should occur, and if so, cell division and molecular partitioning was realized. We note that we tried smaller fixed time steps (3, 1.5 and 0.75 min) and found no difference in the numerical solutions. Though more complex algorithms exist [47,49], in order to speed up our parameter inference, we replaced the SSA with a fixed Tau-leaping algorithm. However, if we encountered a parameter set which produced a negative quantity using Tau-leaping during parameter inference, we would switch to using the SSA. This allowed us to reduce the computation time for parameter inference while not introducing significant errors. We also confirmed that the parameter sets obtained in the final posterior distributions yielded similar results when simulated using the SSA.

**Algorithm 1.** Pseudocode of algorithm used to simulate MGMT dynamics. This algorithm was used to simulate the model defined by chemical reactions (2.10), (2.12)–(2.14), (2.17), (2.18) and (2.27) or (2.28) coupled with ODE solution (2.4) in the absence of TMZ or a time discretization of (2.1) and (2.3) in the presence of TMZ, using parameters defined in tables 1 and 2 and initial conditions in electronic supplementary material, table S1.

(i) Create cell array with $N \times 9$ empty entries, where $N$ corresponds to the maximum number of cells simulated and 9 corresponds to the number of model variables. Set $t = 0$ and for each cell $i$ set $V_i(0) = 15$, $DNA_i(0) = 1$, $DNA_{REF_i}(0) = 1$, $MGMTmRNA_i(0) = 10$, $MGMT_i(0) = 100$, $REFmRNA_i(0) = 10$, $REF_i(0) = 100$, $TMZ(0) = 0$ and compute $V_F$ for each cell. We also assume all cells are living at $t = 0$, i.e. for all $i$ cell$_i(0) =$ true.

(ii) Generate two random numbers $\xi_1$ and $\xi_2$ uniformly distributed in (0,1).

(iii) For each cell $i = 1$ to $N$, evaluate the propensity functions for the following reactions:

If cell$_i(t) = $ true

$$\alpha_{i,1}(t) = b_1 DNA_i(t),$$
$$\alpha_{i,2}(t) = d_1 MGMTmRNA_i(t),$$
$$\alpha_{i,3}(t) = b_2 MGMTmRNA_i(t),$$
$$\alpha_{i,4}(t) = b_1 DNA_{REF_i}(t),$$
$$\alpha_{i,5}(t) = d_1 REFmRNA_i(t),$$
$$\alpha_{i,6}(t) = b_2 REFmRNA_i(t),$$
$$\alpha_{i,7}(t) = \epsilon + \frac{k_d \cdot TMZ(t)}{k_d \cdot TMZ(t) + s \cdot MGMT_i(t)},$$

else

$$\alpha_{i,1}(t) = \alpha_{i,2}(t) = \alpha_{i,3}(t) = \alpha_{i,4}(t) = \alpha_{i,5}(t) = \alpha_{i,6}(t) = \alpha_{i,7}(t) = 0$$

end

Then evaluate $\alpha_0 = \sum_{i=1}^{N} \sum_{j=1}^{7} \alpha_{i,j}(t)$.

(iv) Compute the time when the next reaction takes place as $t + \tau$, where $\tau$ is given by $\tau = \frac{1}{\alpha_0} \ln \left[\frac{1}{\xi_1}\right]$.

(v) Set $(I,J)$ to be the smallest integers satisfying $\sum_{i=1}^{I} \sum_{j=1}^{J} \alpha_{i,j}(t) > \xi_2 \alpha_0$.

(vi) If cell$_i(t) =$ true

If $J = 1$, set $MGMTmRNA_I(t + \tau) = MGMTmRNA_I(t) + 1$.

If $J = 2$, set $MGMTmRNA_I(t + \tau) = MGMTmRNA_I(t) - 1$.

If $J = 3$, set $MGMT_I(t + \tau) = MGMT_I(t) + 1$.

If $J = 4$, set $REFmRNA_I(t + \tau) = REFmRNA_I(t) + 1$.

If $J = 5$, set $REFmRNA_I(t + \tau) = REFmRNA_I(t) - 1$.

If $J = 6$, set $REF_I(t + \tau) = REF_I(t) + 1$.

If $J = 7$, set cell$_I(t + \tau) = $ false

Update propensities.

end

(vii) For all $i$ where cell$_i(t) = $ true, set $V_i(t + \tau) = V_i(t) \exp \left( \frac{\mu_0(t + \tau)}{1 + TMZ(t + \tau)/I_{50}} \right)$ and $TMZ(t + \tau) = f_i(t + \tau)$ where $i = 3,4,5$. Check if $V_i(t + \tau) \geq V_F$ and then binomially distribute contents of cell $i$ between cells $i$ and a randomly selected cell in the population.

(viii) Set $t = t + \tau$, if $t >= t_f$ then end.

## 2.4.3. Modelling TMZ uptake

In order to capture the slow uptake of TMZ into the cells from the medium, we modelled the TMZ intake function as a saturating, sigmoidal function

$$f_3(t) = TMZ \times \frac{t^h}{Q^h + t^h}, \tag{2.29}$$

where $t$ is the time, TMZ is the used TMZ concentration, $h$ is a Hill coefficient which dictates the sharpness of

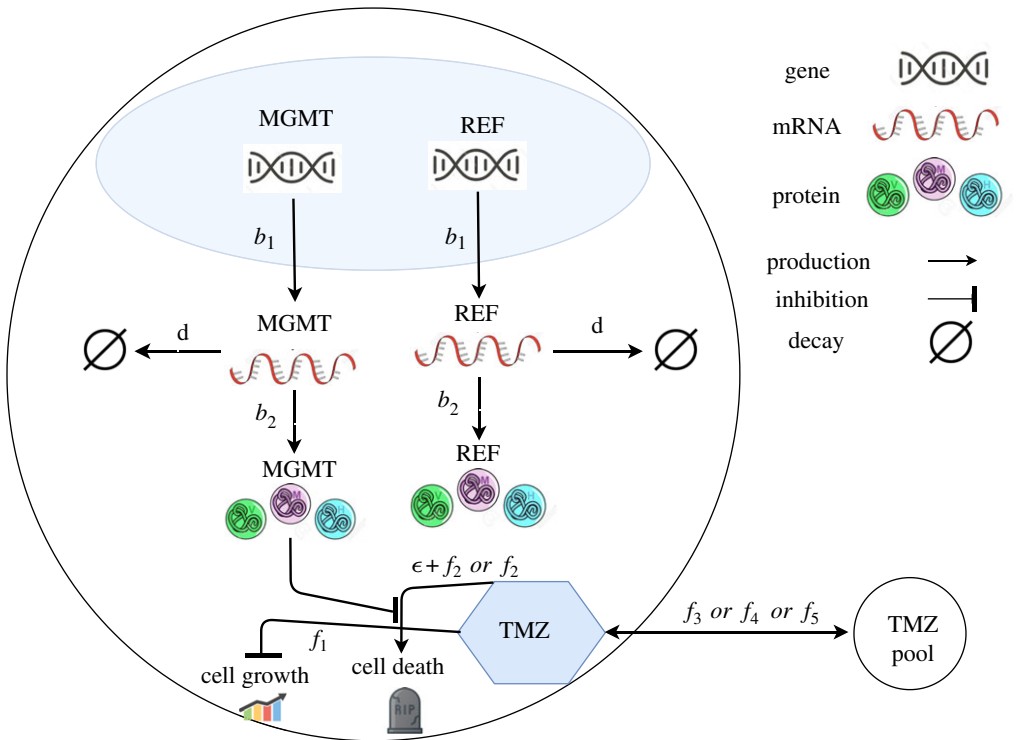

**Figure 1.** Schematic diagram illustrating overall model of MGMT regulation. MGMT is assumed to be constitutively expressed and inhibit the cell death process instigated by TMZ. TMZ is assumed to inhibit cell growth and promote cell death. A reference protein is also modelled in the same way as MGMT but is assumed to play no role in cell death.

TMZ uptake and $Q$ is the time point at which the half-maximal concentration of TMZ enters the cells. We also considered a fast uptake of TMZ into the cells from the medium using the following step function:

$$f_4(t) = \begin{cases} 0, & t \leq t_{in}, \\ [TMZ], & t > t_{in}. \end{cases} \tag{2.30}$$

Finally, we considered a pulse-like term for short TMZ dosing

$$f_5(t) = \begin{cases} 0, & t \leq t_{in}, \\ [TMZ], & t_{in} < t \leq t_{out}, \\ 0, & t > t_{out}, \end{cases} \tag{2.31}$$

where $t_{out}$ indicates when the TMZ administration finishes. For simplicity, we did not model TMZ degradation or dilution.

## 2.5. Parameter estimation

Approximate Bayesian computation (ABC) or likelihood-free methods were developed to deal with models where likelihood calculations fail [50]. Through sampling from unbiased uniform distributions, the ranges of which were inspired by experimental data, we stored parameter sets that yielded simulated data sufficiently resembling the observed data or target data. The ABC approach is found to be comparable to likelihood-based methods that use complete datasets [51]. In this study, we sought to find parameter configurations of our model that yield phenotypic selection, i.e. to answer the question: is there a parameter set that leads to the selective upregulation of MGMT in response to TMZ? To find whether or not our model was capable of exhibiting this phenotypic selection behaviour, we introduced a set of error functions that when minimized produced behaviour consistent with MGMT-mediated drug resistance, i.e. a cell population that stays viable in the presence of TMZ due to an increase in MGMT expression.

—Error$_1$ = ⟨REF⟩/⟨MGMT⟩, where ⟨ ⟩ represents the population mean following TMZ administration. A parameter set that minimizes Error$_1$ will produce a high mean expression of MGMT protein compared to the reference protein.

**Algorithm 2.** ABC algorithm used to estimate the parameter sets that yield phenotypic selection. $N_p$ is the number of particles or parameter vectors stored; here we considered $N_p = 10\,000$. $M_m$ stands for the multi-scale model defined by reactions (2.10), (2.12)–(2.14), (2.17), (2.18) and (2.28), and TMZ uptake modelled by $f_4$ with $t_{in} = 144$ h (resp. (2.10), (2.12)–(2.14), (2.17), (2.18) and (2.27), and TMZ uptake modelled by $f_3$, where $Q = 77.6434$ h and $h = 9.5362$) and ODE solution (2.4) in the absence of TMZ or a time discretization of (2.1) and (2.3) in the presence of TMZ, $\pi(\theta)$ refers to the prior distributions for parameters $\theta$, $\rho(.,.)$ refers to the Euclidean distance, while $S$ is the summary statistics or the Error. In order to estimate parameters in table 1 (resp. table 2), we set Error = Error$_1$ + Error$_2$ (resp. Error = Error$_1$ + Error$_2$ + Error$_3$).

 (i) Initialize uniform parameter prior distributions with ranges from tables 1 and 2, data in vector $y$ and set $\gamma$, the error threshold.

 (ii) while i < $N_p$

 Generate $\theta^* \sim \pi(\theta)$

 Simulate $x \sim M_m(x|\theta^*)$

 if $\rho(S(x),S(y)) < \gamma$

 Store $\theta_i \leftarrow \theta^*$

 $i = i + 1$

 end

 end

—Error$_2$ = $|1 - N(t_f)|$, where $N(t_f)$ stands for the fraction of living cells at the final time point ($t_f$). Minimizing Error$_2$ ensures that some fraction of the cells are living by the end of the simulation.

—We defined Error$_3$ as the mean square error of our model output cell viability time series and the results represented in figure 4*b* for TMZ = 0 and 800 μM. Minimizing Error$_3$ ensures that our model replicates the response of the N15-0385 cell line to TMZ.

The whole set of error functions (resp. Error$_1$ and Error$_2$) was used to calibrate the data-constrained model using experimental data (resp. estimate parameters for the general model). Above we describe the main steps of the ABC algorithm used to estimate the parameters.

Table 1 contains a description of the parameters in our general model, their symbols, the prior ranges used for parameter inference as well as the modes of the posterior distributions found through parameter inference using error functions Error$_1$ and Error$_2$. Table 2 contains a description of the parameters in our data-constrained model, their symbols, the prior ranges used for parameter inference as well as the modes of the posterior distributions found through parameter inference using error functions Error$_1$, Error$_2$ and Error$_3$.

## 2.6. Statistical measures

Noise can be defined using the Fano factor or the coefficient of variation squared which measures the dispersion of the distribution and we used the coefficient of variation squared as follows:

$$\text{noise}[X] = \frac{\sigma^2}{\lambda^2}, \tag{2.32}$$

where $\sigma^2$ and $\lambda$ represent, respectively, the variance and the mean across the cell population of variable X. Skewness is a measure of the asymmetry of a distribution. A distribution is said to be positively skewed when the right tail is longer, i.e the mass of the distribution is concentrated on the left. Mathematically, we define skewness of variable X in the following way:

$$\text{skew}[X] = \mathbf{E}\left[\left(\frac{X - \lambda}{\sigma}\right)^3\right]. \tag{2.33}$$

**Table 1.** The value column refers to the mode (best fit) of the posterior distribution (see electronic supplementary material, figure S3), while the range column refers to the prior distribution of the parameters.

| parameter description | symbol | value | range | units |
|---|---|---|---|---|
| growth rate | $\mu_0$ | 0.0948 | $(e^{-10}, e^1)$ | $10^{-13}\,l\,h^{-1}$ |
| transcription rate | $b_1$ | 0.0016 | $(e^{-10}, e^1)$ | $h^{-1}$ |
| mRNA decay | $d$ | 0.0003 | $(e^{-10}, 1)$ | $h^{-1}$ |
| MGMT de-alkylation rate | $s$ | 202.25 | $(e^{-10}, e^{10})$ | $\mu M^{-1}$ |
| half-inhibition concentration | $I_{50}$ | 0.0013 | $(e^{-10}, e^{10})$ | $\mu M$ |
| translation rate | $b_2$ | 5483.75 | $(e^{-10}, e^{10})$ | $h^{-1}$ |
| basal cell death | $\epsilon$ | 0.0 | — | $h^{-1}$ |
| TMZ-mediated cell death | $k_d$ | 1.92 | $(e^{-10}, e^{10})$ | $\mu M\,h^{-1}$ |
| final volume standard deviation | $\sigma_1$ | 0.2 | — | $10^{-13}\,l$ |
| division standard deviation | $\sigma_2$ | 0.05 | — | dimensionless |
| initial volume of the cell | $V_I$ | 15 | — | $10^{-13}\,l$ |
| noisy linear map slope | $a$ | 1.0 | — | dimensionless |
| noisy linear map intercept | $b$ | 15.0 | — | $10^{-13}\,l$ |

**Table 2.** The value column refers to the mode (best fit) of the posterior distribution (see electronic supplementary material, figure S5), while range column refers to the prior distribution of the parameters.

| parameter description | symbol | value | range | units |
|---|---|---|---|---|
| growth rate | $\mu_0$ | 0.01388 | — | $10^{-13}\,l\,h^{-1}$ |
| transcription rate | $b_1$ | 0.00882 | (0,1) | $h^{-1}$ |
| mRNA decay | $d$ | 0.07847 | (0,0.1) | $h^{-1}$ |
| MGMT de-alkylation rate | $s$ | 96.8167 | (0,100) | $\mu M^{-1}$ |
| half-inhibition concentration | $I_{50}$ | 0.4912 | (0,1) | $\mu M$ |
| translation rate | $b_2$ | 9.1027 | (0,100) | $h^{-1}$ |
| basal cell death | $\epsilon$ | 0.00043 | (0,0.1) | $h^{-1}$ |
| TMZ-mediated cell death | $k_d$ | 0.1808 | (0,10) | $\mu M\,h^{-1}$ |
| time point of half-maximal concentration | $Q$ | 77.6434 | — | $h$ |
| Hill coefficient | $h$ | 9.5362 | — | dimensionless |
| final volume standard deviation | $\sigma_1$ | 0.2 | — | $10^{-13}\,l$ |
| division standard deviation | $\sigma_2$ | 0.05 | — | dimensionless |
| initial volume of the cell | $V_I$ | 15 | — | $10^{-13}\,l$ |
| noisy linear map slope | $a$ | 1.0 | — | dimensionless |
| noisy linear map intercept | $b$ | 15.0 | — | $10^{-13}\,l$ |

# 3. Results

## 3.1. Cell death-driven phenotypic selection

Our multi-scale model of MGMT dynamics couples constitutive gene expression of a DNA-REPAIR protein MGMT to cell growth, death, division and partitioning. Each cell expresses MGMT which we assumed negatively impacts the cell death rate due to TMZ (through the process of de-alkylation). We also modelled a reference protein expressed at the same rate which we call 'REF' as a control. An ABC algorithm was used to estimate the parameters that yielded phenotypic selection (see Methods, Algorithm 2). The best-fit results are summarized in table 1 and initial conditions used for simulations are presented in electronic

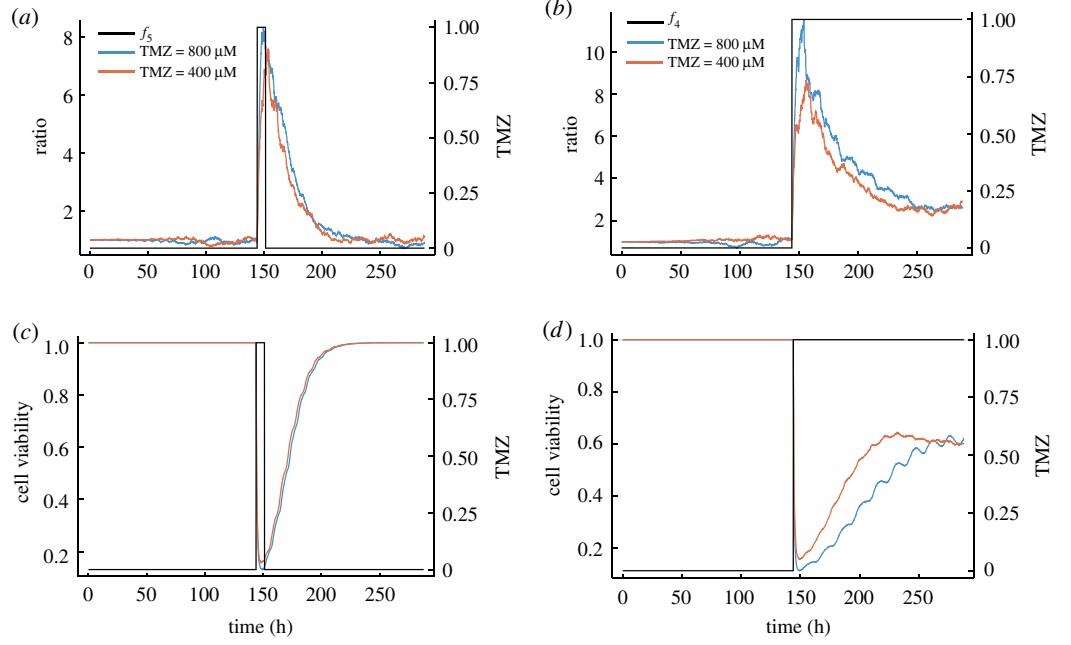

**Figure 2.** Dose-dependant, reversible upregulation of MGMT. Plots show simulations from model defined by chemical reactions (2.10), (2.12)–(2.14), (2.17), (2.18) and (2.28) coupled with ODE solution (2.4) in the absence of TMZ or a time discretization of (2.1) and (2.3) in the presence of TMZ, using parameters defined in table 1 and initial conditions in electronic supplementary material, table S1 and TMZ uptake modelled by $f_4$ with $t_{in} = 144$ h for (*b,d*) and $f_5$ with $t_{in} = 144$ h and $t_{out} = 150$ h for (*a,c*). (*a,b*) The ratio = $\langle MGMT \rangle / \langle REF \rangle$ in response to different regimes (transient and constant) and doses of TMZ (400 and 800 μM). (*c,d*) The cell viability in response to different regimes (transient and constant) and doses of TMZ (400 and 800 μM).

supplementary material, table S1. Using the best-fit parameter set, we simulated the model response to different TMZ doses and compared the MGMT and REF levels. Example simulations are shown in figure 2. We defined the ratio (or the phenotypic selection) at time $t$, as $\langle MGMT(t) \rangle / \langle REF(t) \rangle$, where $\langle \ \rangle$ is the mean value computed over the cell population at time $t$, i.e. a ratio = 1 means that MGMT and REF are expressed at similar levels, while a ratio > 1 means that MGMT is overexpressed compared to REF. We used a simulation time of 288 h, which is twice that of our experimental cell viability assay time, and the TMZ intake function is modelled as a step function ($f_4$ with $t_{in} = 144$ h), as in the upper panel of figure 2*b*, which assumes that cells are instantly submerged in TMZ. In a later section, we relaxed this assumption and used a sigmoidal function, $f_3$, to capture the slow diffusion and uptake of TMZ into individual cells. Our general model shows in figure 2*a* a selective, reversible upregulation of MGMT in response to transient TMZ ($f_5$ with $t_{in} = 144$ h and $t_{out} = 150$ h) in a dose-dependent manner. We found that, in the presence of constant TMZ, it was not possible to observe the same complete reversibility of the cell viability; instead, as can be seen in figure 2*b*, we observe a selective, partially reversible upregulation of MGMT in response to constant TMZ in a dose-dependent manner. We note that though the MGMT phenotypic selection is reversible, MGMT returns to the initial amount on a timescale which is consistent with the cell doubling time (which can be long for glioma cells). Hence, even following a transient dose of TMZ, subsequent generations of GBM cells can be protected due to elevated levels of MGMT. This is because the cell death rate is inversely proportional to the level of MGMT. Hence, if we were to administer another dose of TMZ at 160 h, this would be less effective in killing the cells than administering another at 250 h. In figure 2*c,d*, we show the ability of our general model to exhibit drug resistance where we plot the cell viability in response to different TMZ regimes and doses. Following sustained TMZ administration, the simulation shows a recovery and non-zero steady-state behaviour (at about 60% of total population), suggesting that the cell population has balanced cell birth and death processes due to MGMT expression levels. We note that this recovery in the presence of sustained TMZ (figure 2*d*) is slower and smaller than the case where TMZ is only transiently administered (figure 2*c*). In figures 2*b,d*, we observed that acquired phenotypic resistance occurs in a stable manner as long as TMZ is administered with the oscillatory behaviour indicating repeated cell division and death events. We also continued our simulation for up to 1000 h to verify a steady state was reached.

## 3.2. Statistical properties of phenotypic selection

In order to understand the parametric requirements for phenotypic selection, a correlation analysis was performed using the top 100 best-fit parameter sets from the ABC parameter estimation. The correlation analysis results are presented in electronic supplementary material, figure S3. We find that a higher cell growth and TMZ-mediated cell death rate favour phenotypic selection. In general, we note that we may need a strong positive correlation between $k_d$ and $s$ as well as a positive correlation between $\mu$ and $I_{50}$, in order to observe phenotypic selection. These relationships are consistent with intuition, as if $k_d$ is high this means that TMZ effectively kills cells, therefore, we would need a high de-alkylation rate to repair the damage caused by TMZ. On the other hand, if the cells grow faster, i.e, $\mu$ is high, TMZ must be able to inhibit cell growth, i.e, high $I_{50}$. We also found that in order to observe phenotypic selection, a weak negative correlation may be needed between $b_2$ and $s$, $I_{50}$ and $b_1$, and $I_{50}$ and $s$. We also observe that our model is poorly constrained or sloppy in some parameter directions [52], for example $I_{50}$ and $d$, where model output seems robust to different combinations of these parameters. Interestingly, we note that the parameter $I_{50}$ appears bimodal and can maintain a low error for either high or low values—which may correspond to the trade-off between minimizing Error$_1$ and Error$_2$. Furthermore, based on parameter sensitivity analysis presented in electronic supplementary material, figure S4, we found that the growth rate $\mu$, the translation rate $b_2$ and the transcription rate $b_1$ are the most sensitive, which implies that gene expression and cell growth have to be tuned precisely to lower both Error$_1$ and Error$_2$ simultaneously.

We noted that a lower transcription rate was associated with a higher ratio of MGMT to the reference protein, this result was confirmed by a sensitivity analysis presented in electronic supplementary material, figure S4, where we show that the parameter $b_1$ corresponding to the transcription rate is sensitive. This suggests that in the presence of TMZ a high level of mRNA noise is required for phenotypic selection. In order to confirm this we manually varied the level of the mRNA noise and observed the impact on the ratio of MGMT to REF protein. Since we used a constitutive gene expression model (see Methods section), the transcription of both MGMT mRNA and REF mRNA are simple birth–death processes. The noise can be written in the case of a birth–death process (leading to a Poisson term [53]) as

$$\text{noise}[X] = \frac{1}{\lambda}. \tag{3.1}$$

To further probe the statistical properties of phenotypic selection, we also varied the skewness of the mRNA distribution. In the case of a birth–death process, the skewness can be written,

$$\text{skew}[X] = \lambda^{-\frac{1}{2}}, \tag{3.2}$$

where $\lambda$ is the mean. We used the noise and skewness definitions as defined in equations (3.1) and (3.2). It is possible to construct a mean field steady-state approximation of the model as in [54], and by manipulating the resultant equations we found that $\langle \text{mRNA} \rangle = b_1/(d+\mu)$ and $\langle \text{MGMT} \rangle = b_1 \cdot b_2/(d+\mu)\mu$. Thus, if we varied the mRNA levels by decreasing parameter $b_1$ while increasing the parameter $b_2$ by the same amount, we were able to manipulate the mean steady-state mRNA levels whilst maintaining the mean steady-state protein levels constant and thus vary the mRNA noise and skewness. In figure 3, random bias in partitioning was omitted, i.e. $\eta_1 = \eta_2 = 0$, and we used the best-fit parameter set as a baseline and then show the result of decreasing $b_1$ while increasing $b_2$. Our findings presented in figure 3$a$ reveal a striking dependence between the mRNA noise and level of phenotypic selection. In particular, we found the existence of a positive relationship between the noise, skewness (see figure 3$b$) and the ratio of the mean levels of MGMT to the mean levels of reference protein, suggesting that a high noise and skewness in mRNA levels are needed for phenotypic selection in the presence of TMZ (figure 3$a$,$b$). We note that in the absence of TMZ neither the skewness or noise of mRNA affected the ratio of the mean levels of MGMT to the mean levels of reference protein. Since protein noise is dominated by mRNA noise [55], the constant levels of TMZ remove cells with low values of MGMT from the population and only allow those with high MGMT levels to divide and multiply, becoming more represented in the population. This is reflected in figure 3$c$,$d$ where we plotted the cell viability recovery. We found that increasing the noise in gene expression causes higher levels of recovery. On the contrary, if the cell population had the same homogeneous levels of protein (i.e. low noise levels), the whole population would respond the same way to TMZ treatment and there would either be no cell death or no recovery.

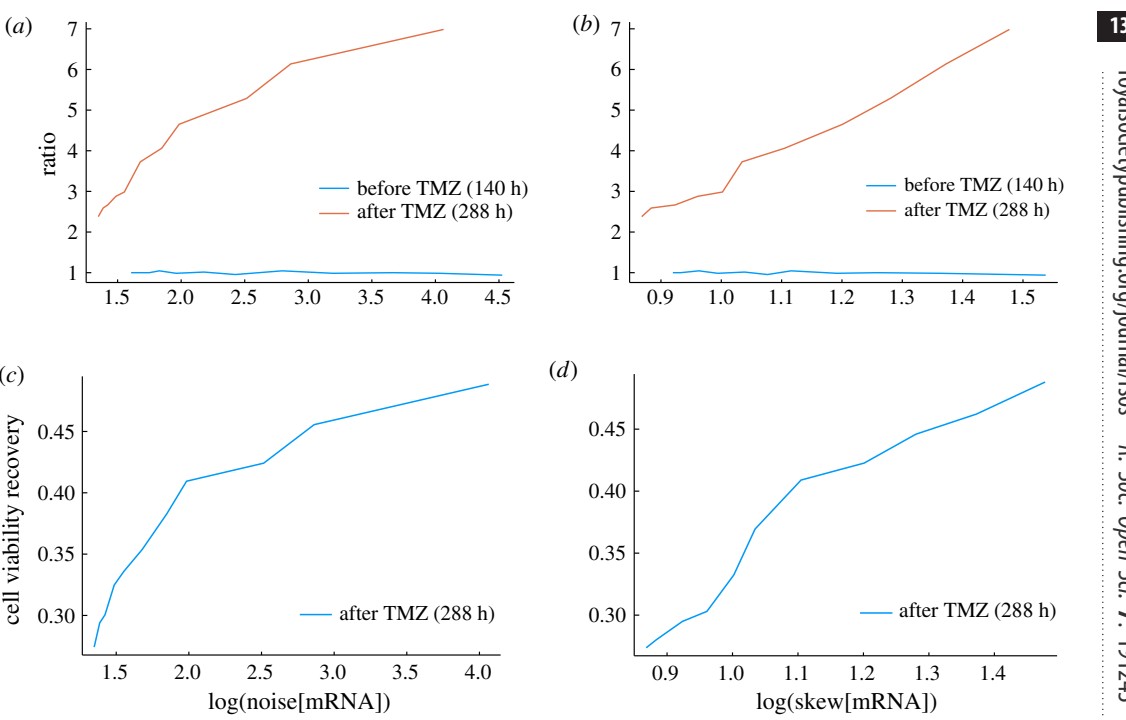

**Figure 3.** High mRNA noise and skewness are associated with phenotypic selection in the presence of TMZ. Plots show simulations from model defined by chemical reactions (2.10), (2.12)–(2.14), (2.17), (2.18) and (2.28) coupled with ODE solution (2.4) in the absence of TMZ or a time discretization of (2.1) and (2.3) in the presence of TMZ, using parameters defined in table 1 and initial conditions in electronic supplementary material, table S1 and TMZ uptake modelled by $f_4$ with $t_{in} = 144$ h. Plots (a) and (b) show how the ratio defined by $\langle MGMT \rangle / \langle REF \rangle$ varies with noise and skewness of mRNA presented on a logarithmic scale. Plots (c) and (d) show cell viability recovery following TMZ administration versus the noise and skewness of mRNA presented on a logarithmic scale.

## 3.3. GBM patient-derived cells exhibit a modest response to TMZ

In the previous two sections, we presented results which explored our model for general unconstrained parameter sets. Our motivation for this was so that we could uncover general parameter relationships necessary for phenotypic selection. We next sought to push our model into more realistic parameter regimes. To determine the doubling time of N15-0385 GBM patient-derived cell line, cells were counted every day for 6 days and live cell numbers were used to generate the cell growth curve presented in figure 4a. The doubling time was obtained using an ABC algorithm defined in the Methods section. We used the mean square error of the exponential growth function and three replicates of the experimental data rather than taking the mean of the replicates. This allowed us to capture the variability of the data and find a value which was truly reflective of the cell growth dynamics. For this cell line, we found the doubling time to be 49.956 h, which is consistent with the literature where the doubling time for glioma cells *in vitro* has been reported to vary between 40 and 60 h [56].

To investigate the potential of TMZ to induce cell death in GBM cells, we assessed cell viability after treatment. Treatment of the cells with TMZ resulted in cell death in a dose- and time-dependent manner. N15-0385 cells appear to be resistant to TMZ treatment within clinically achievable concentrations even 144 h post-treatment and only a modest response was found using higher concentrations 96–144 h post-treatment, as shown in figure 4b. Following these findings, we then used our model to understand possible mechanisms behind this modest response. In particular, we wanted to assess if our model could exhibit a selective upregulation of MGMT with the cell death parameters constrained using this relevant clinical data. Notably, from our original exploration of our model, we found that in order to maximize phenotypic selection, the TMZ-mediated cell death parameter $k_d$ had to be high, it was not clear whether this was essential to observe a selective upregulation of MGMT.

## 3.4. Model calibration and validation using experimental data

To calibrate our model to the experimental data presented in figure 4, we introduced a couple of modifications to our model. We fixed the growth rate to the best-fit parameter discussed in the previous

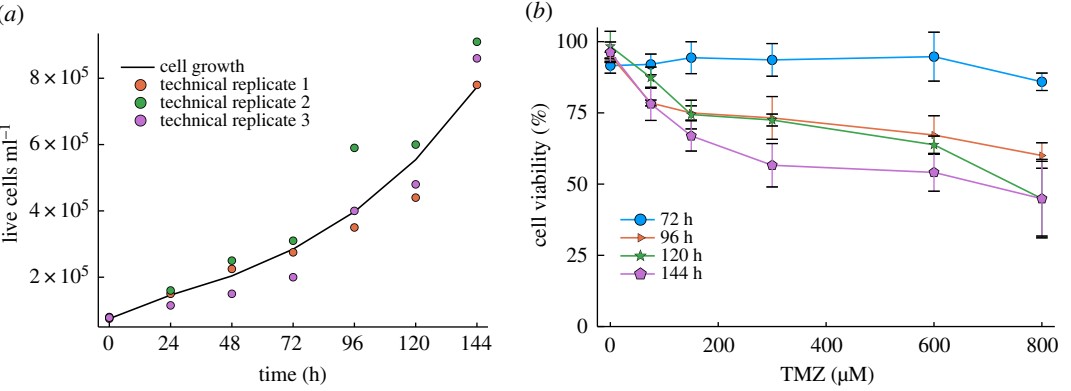

**Figure 4.** Doubling time and TMZ responsiveness of the GBM patient-derived cell line N15-0385. (a) The cell proliferation data of GBM patient-derived cell line N15-0385 was determined from live cell numbers counted over a 6-day period. The line represents the result of fitting the exponential growth function to the data. (b) Cell survival was measured following treatment with increasing concentrations of TMZ (0–800 μM) at 72, 96, 120 and 144 h post-treatment. Data show cell survival relative to control values of 100%. For both experiments, data are expressed as mean ± s.e.m. of three independent experiments performed in triplicate.

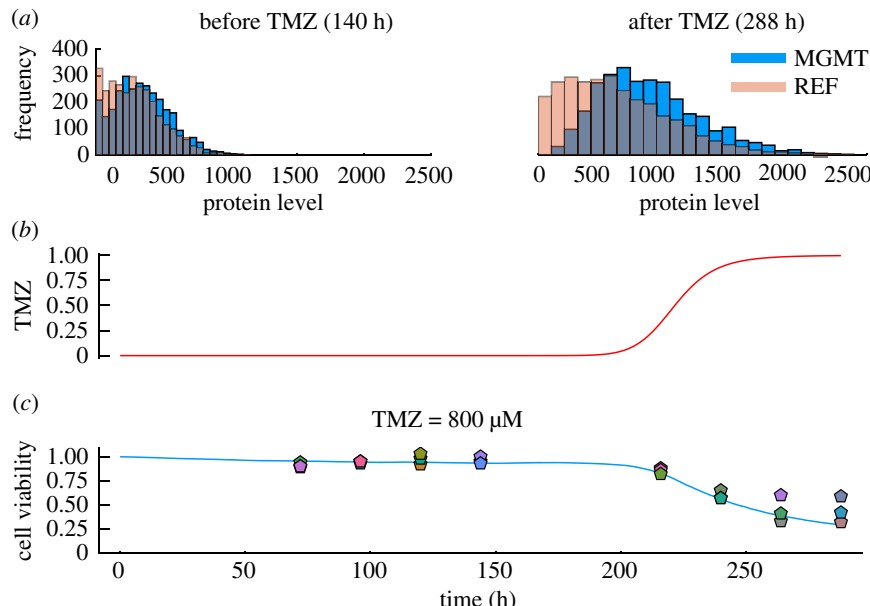

**Figure 5.** Model calibration for TMZ = 0 and 800 μM, MGMT and REF distributions before and after TMZ administration. Plots show simulations from model defined by chemical reactions (2.10), (2.12)–(2.14), (2.17), (2.18) and (2.27) coupled with ODE solution (2.4) in the absence of TMZ or a time discretization of (2.1) and (2.3) in the presence of TMZ, using parameters defined in table 2 and initial conditions in electronic supplementary material, table S1. Plot (a) shows MGMT and REF distributions before and after TMZ administration for the cell population. Plot (b) represents the TMZ intake function modelled by $f_3$, where $Q = 77.6434$ h and $h = 9.5362$. Plot (c) shows the cells survival percentage in response to TMZ. The pentagons represent the technical triplicates at each time point.

section and introduced a new parameter, $\epsilon$ to represent background cell death. Furthermore, we relaxed the assumption regarding the instant infusion of TMZ to steady-state levels and introduced a sigmoidal TMZ intake function defined in equation (2.29). We then used ABC to fit the cell viability data while maximizing phenotypic selection for TMZ = 0 μM and TMZ = 800 μM cell viability cases. The best-fit results are summarized in table 2. We found that the model can closely replicate the observed experimental response to TMZ = 800 μM, as shown in figure 5c while simultaneously producing a selective upregulation of MGMT, implying that even with clinically relevant cell death rates, our model can exhibit phenotypic selection. To probe this further, we compared the protein distributions of MGMT and the reference protein as depicted in figure 5a. Prior to TMZ administration, MGMT and REF have a

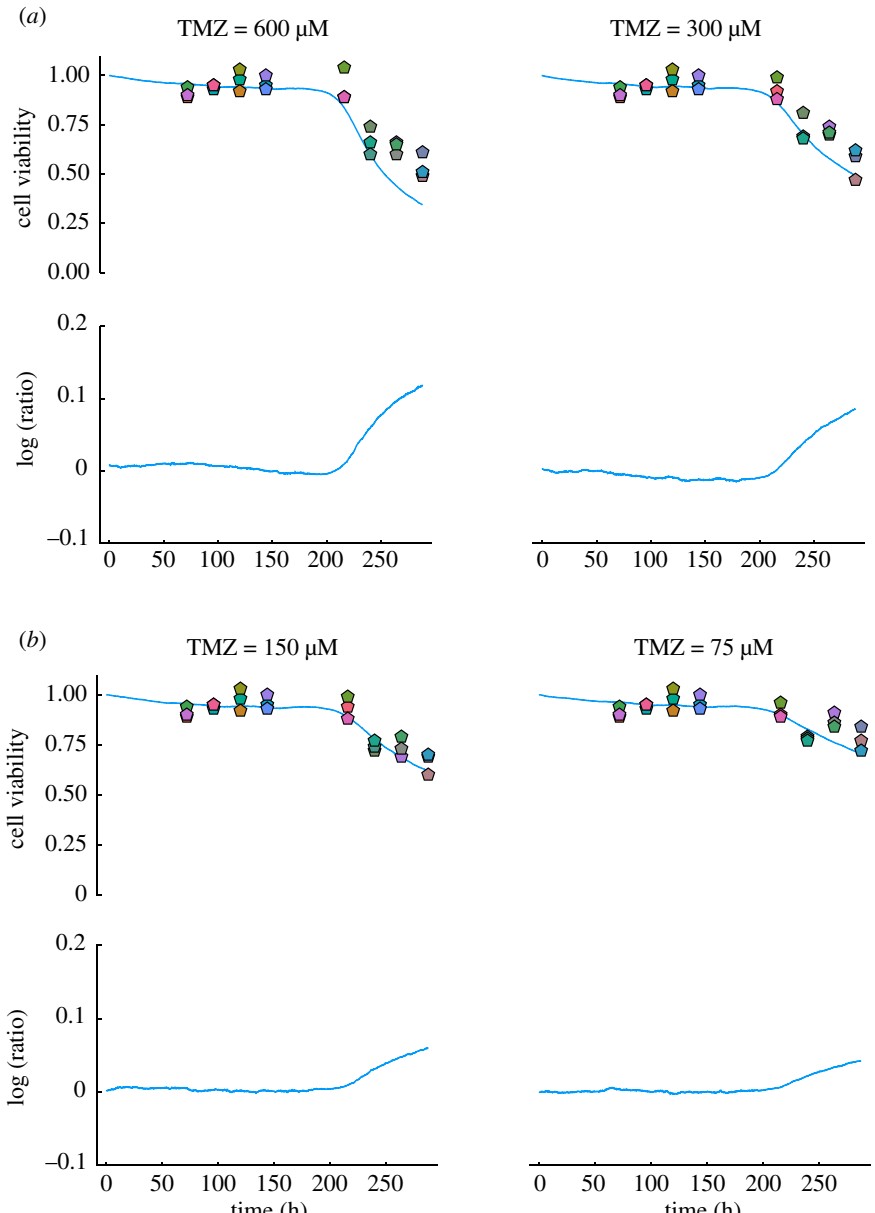

**Figure 6.** Model validation using cell viability results. Panels show cell viability and log(ratio) where ratio is defined by $\langle MGMT \rangle / \langle REF \rangle$ simulations in response to different TMZ doses following $f_3$ where $Q = 77.6434$ h and $h = 9.5362$, from model defined by chemical reactions (2.10), (2.12)–(2.14), (2.17), (2.18) and (2.27) coupled with ODE solution (2.4) in absence of TMZ or a time discretization of (2.1) and (2.3) in the presence of TMZ, using parameters defined in table 2 and initial conditions in electronic supplementary material, table S1. The pentagons represent the technical triplicates at each time point. The ratio is presented on a logarithmic scale. The cell viability simulation results are shown for TMZ = 600, 300, 150 and 75 $\mu$M.

similar distribution and most of the cells have a protein level lower than 500 molecules. After TMZ administration, we notice a unimodal shift to the right of the MGMT distribution, this suggests that the cells with high MGMT levels were selected and survived while those with low MGMT levels died. Once calibrated, we used the model to simulate the long-term outcome of the cell population and found that even with the use of TMZ = 800 $\mu$M a few cells survived due to this phenotypic selection effect. We observed that these few remaining cells accumulated very high MGMT levels and were extremely resistant. We also simulated the impact of stopping TMZ administration and found that these few remaining resistant cells begin to divide and grow (data not shown) implying that this mechanism may be responsible for the poor prognosis of glioma patients.

Next, we found that our model can successfully capture additional data. The model response to TMZ = 75, 150, 300 and 600 $\mu$M was simulated and the results are presented in figure 6. Overall, the

calibrated model can accurately predict the cell survival rate in response to TMZ = 75, 150 and 300 μM. We note that our model failed to accurately capture the data in the case of TMZ = 600 μM, though the data appear to exhibit unusually high variance for early time points. Once again, we observed a dose-dependence of phenotypic selection, i.e. as we increased the TMZ dose, the ratio of MGMT to the reference protein increased (displayed in 2nd and 4th rows of figure 6).

Finally, we studied the parameter sets that provided the best fit to the cell viability data while maximizing phenotypic selection (see electronic supplementary material, figure S5). As opposed to the general case where we put little constraints on the uniform priors used (see electronic supplementary material, figure S3), here we attempted to limit the parameter space to more realistic regimes and so limited the prior ranges to more modest ranges. As in the general case, we found that the transcription rate had to be very small (to introduce sufficient mRNA noise) for the selective upregulation of MGMT. Unlike the general case, we observed a much smaller range for the posterior distribution of the cell death parameter $k_d$, which is constrained tightly by the cell viability data. A significant strong positive correlation exists between the translation rate $b_2$ and the mRNA decay rate $d$ as well as a positive correlation between $k_d$ and $s$ (as observed in electronic supplementary material, figure S3). We also note that the background cell death parameter is the most sensitive to change followed by the mRNA transcription rate. Full sensitivity details can be seen in electronic supplementary material, figure S6.

# 4. Discussion and conclusion

Drug resistance in GBM remains an unsolved problem. Many studies have implicated that MGMT expression levels play a role in treatment outcome. In this study, a stochastic multi-scale model was developed to understand the potential role of MGMT expression in the modest response of GBM cells to TMZ. Without any assumptions regarding mutations or complex gene regulatory networks and instead using minimal biologically realistic assumptions, our model reveals phenotypic selection as a possible explanation of drug resistance development in GBM. Our mathematical model was capable of producing a selective upregulation of MGMT in response to TMZ coupled with a recovery of the cell population, suggesting that cells with high MGMT levels divided and survived, i.e. phenotypic selection through cell death. By taking this microscopic description of MGMT expression and coupling it to a model of cell growth, division and death, we were able to dig into the details of what could drive phenotypic selection of MGMT and subsequently impede TMZ-mediated cell death. All components of the model are necessary (except for basal cell death, which was only needed for capturing our cell viability data) to observe the phenotypic selection behaviour. If we were to exclude any components of the model, for example, cell growth, then we would lose the recovery of the cell population following TMZ administration, while if we eliminated cell death, we would lose the selection of MGMT-rich cells.

The model we developed consists of two levels, the intracellular and cell population levels. For the intracellular processes, we used the stochastic simulation or Tau-leaping algorithms to simulate the canonical stochastic gene expression reactions (transcription, mRNA decay and translation). Initially, we considered two different models of stochastic gene expression, a simple constitutive gene expression model which we presented in this paper (figure 1) and a gene switching model as in [57,58]; the latter being identical to the model presented here in every reaction but the gene state could now switch between 'on' and 'off' states. A Bayesian model selection using error functions Error$_1$ and Error$_2$ revealed that the constitutive gene expression model was more probable (with probability, $p = 0.55$) so we adopted this for our study though we note that the same results could be presented for the gene switching model of stochastic gene expression (though it used more parameters). We coupled these intracellular reactions based on constitutive gene expression to an exponential growth differential equation to simulate cell growth and implemented rules to simulate cell division, partitioning and cell death.

In order to model the impact of TMZ on GBM cells, we also defined drug-related parameters $I_{50}$ and $k_d$ which respectively describe the half inhibition of growth rate concentration of TMZ and TMZ-mediated cell death rate. These parameters are critical for observing phenotypic selection and cell viability recovery. In particular, we found that if $k_d$ was decreased sufficiently (while keeping all other parameters at the baseline defined in table 1) both phenotypic selection and cell death recovery were lost (electronic supplementary material, figure S7(A)). This confirmed that phenotypic selection is driven by stochastic gene expression and cell death alone. This is consistent with the findings of

[25,27] while other studies have shown other non-genetic mechanisms of phenotypic selection such as biased cell partitioning or stochastic switching [59,60]. Our results are also consistent with Kitange *et al.* where it was shown that MGMT protein expression in cell lines derived from GBM xenografts increased following TMZ administration [12]. Furthermore, based on our cell viability recovery results, we can extrapolate that patients with high protein expression of MGMT would respond poorly to TMZ treatment, which is consistent with [61]. This high protein expression was reflected in the parameter distributions we found to be associated with phenotypic selection, where we found a large translation rate was necessary.

Moreover, we used parameter $s$ to represent the de-alkylation rate which can be thought of as a measure of the efficacy of MGMT in repairing TMZ-induced damage in DNA. We found that when this parameter was decreased (while keeping other parameters at baseline in table 1), the ratio of MGMT to the reference protein increased (electronic supplementary material, figure S7(B)), though if it was made exactly equal to zero, we lost any phenotypic selection effect (electronic supplementary material, figure S8). Whereas if we increased the de-alkylation rate (with respect to the baseline parameter value), we found that phenotypic selection was smaller but the cell viability recovery was larger. This suggests that as long as the de-alkylation rate is sufficiently high, the cells will develop resistance to TMZ. We also varied this parameter to verify our pseudo-steady-state approximation (which we used to reduce the complexity of our model). As shown in electronic supplementary material, figure S7, this approximation is in good agreement with the full model (without steady-state approximation, i.e. reactions (2.8)–(2.18)) for a large range of $s$ values. In particular, we found that as long as $s$ was sufficiently large, the agreement was good and the values of $s$ found to be consistent with phenotypic selection were typically large (see electronic supplementary material, figures S3 and S5).

We studied in detail the parameter sets that yielded phenotypic selection (see electronic supplementary material, figures S3 and S5) and found that a low transcription rate is necessary in order to observe phenotypic selection. This suggests that in the presence of TMZ, a high level of mRNA noise is required for phenotypic selection. We confirmed this by manually varying the model parameters so that the mRNA noise levels changed while keeping the protein levels constant. We found mRNA noise levels showed an almost linear relationship with the amount of selective upregulation of MGMT in the presence of TMZ. Additionally, we found a positive relationship between the level of skewness in the mRNA distribution and the level of selective upregulation of MGMT in the presence of TMZ. In order to test the robustness of our model and understand the relationships between the parameters and the simulation results, we performed a sensitivity analysis for the general model (see electronic supplementary material, figure S4) and the data-constrained model (see electronic supplementary material, figure S6) by inverting the covariance matrix of the final posterior distribution obtained from ABC fitting for each model [62]. In the case of the general model, we found that the translation rate, growth rate and the transcription rate are the most sensitive parameters, while TMZ growth rate inhibition is the least sensitive. In the case of the data-constrained model, we found that the basal cell death rate is the most sensitive followed by the transcription rate, reinforcing the important role that cell death and mRNA noise play in phenotypic selection.

We used experimental data (electronic supplementary material, files S1 and S2) and an ABC algorithm to constrain our model parameters so that the cell death and growth rates were realistic while also seeking parameters that yield phenotypic selection. We found that our model was still able to produce phenotypic selection even with these additional constraints. After calibrating our model to the N15-0385 cell line response to TMZ using two different doses, we found it was also able to successfully predict the response to intermediate doses in all but one case. However, we note that even in the data-constrained case some parameters were left unconstrained (such as $d$, $s$, $I_{50}$ and $b_2$). This implies that we would need additional data to constrain these parameters. One limitation of our study was the lack of gene expression data, this would undoubtedly help constrain more parameters.

Our modelling suggests a potential novel strategy for overcoming MGMT-mediated TMZ drug resistance. As shown in figure 3, we find that we only observe a selective upregulation of MGMT if there exists sufficient noise at the mRNA level. It is known that negative feedback of gene expression at the mRNA level can decrease noise in mRNA levels [63]. Hence, by introducing a negative feedback at the transcription level of MGMT, we could decrease mRNA noise and potentially induce a more pronounced cell death response to TMZ.

Finally, we found that including active protein degradation for both the MGMT and reference proteins had no impact on our results for a large range of degradation rates (see electronic supplementary material, figure S1). Furthermore, it is known that MGMT acts as a suicide enzyme and is therefore consumed in the process of de-alkylation. Hence, we included an additional

degradation term for MGMT alone. The rate of this degradation was computed to be the same as the de-alkylation rate (electronic supplementary material, equation S5). Remarkably, we found that even with this degradation term solely impacting MGMT levels, we were still able to observe a phenotypic selection of MGMT and cell viability recovery (see electronic supplementary material, figure S2A). Moreover, we found our initial results presented in electronic supplementary material, figure 2b could be recovered exactly by decreasing the TMZ-mediated cell death rate (electronic supplementary material, figure S2B). We note though that due to limited computational resources we did not redo parameters estimation when accounting for MGMT and reference proteins degradation and MGMT suicide enzymatic activity.

While we could not find evidence of other mathematical models of MGMT regulation, Storey *et al.* recently developed a mathematical model which focused on the methylation status of MGMT. The authors reported that their model indicated a potential link between TMZ and methylation maintenance [22]. In future work, we aim to investigate a more holistic mathematical model of MGMT regulation in the case of GBM, examine DNA damage contribution to TMZ-induced drug resistance and identify whether the various genetic and epigenetic signalling pathways responsible for MGMT regulation impact our findings reported in this study, which used a reductionist modelling approach. We focused mainly on intrinsic noise in this work, though recent studies have made it clear that extrinsic noise can play a crucial role in the dynamics of intracellular signalling pathways [64–66] and we plan to also explore the impact of extrinsic noise in future work.

Data accessibility. Data supporting this paper are contained within the paper and the electronic supplementary material.
Authors' contributions. V.J., M.V., F.B., A.I. and B.M. contributed to the design and performance of experimental studies. A.L., A.K. and M.S. designed and performed modelling experiments. A.L. produced figures. All authors contributed to manuscript writing. All authors reviewed and approved manuscript prior to submission.
Competing interests. The authors declare that they have no competing interests.
Funding. This work was supported by the European Union's Horizon 2020 research and innovation programme under the Marie Skłodowska-Curie ITN initiative (grant no. 766069).
Acknowledgements. The authors wish to acknowledge the DJEI/DES/SFI/HEA Irish Centre for High-End Computing (ICHEC) for the provision of computational facilities and support.

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
