## [Reviewer comments · Royal Society Open Science]

Review History

RSOS-191243.R0 (Original submission)

Review form: Reviewer 1

Is the manuscript scientifically sound in its present form?

No

Are the interpretations and conclusions justified by the results?

No

Is the language acceptable?

Yes

Do you have any ethical concerns with this paper?

No

Have you any concerns about statistical analyses in this paper?

No

Recommendation?

Major revision is needed (please make suggestions in comments)

Comments to the Author(s)

As a general comment, if the idea underlying the modelling approach is very interesting, the manuscript lacks clarity and precision, both in regards of scientific aspects and accessibility to the reader. Therefore, I recommend a major revision and I strongly suggest you rewrite the manuscript to make it suitable for review (asking for an external reading before submitting the manuscript would be a good idea). Please find in attached file (Appendix A) all my detailed comments.

Review form: Reviewer 2

Is the manuscript scientifically sound in its present form?

Yes

Are the interpretations and conclusions justified by the results?

Yes

Is the language acceptable?

Yes

Do you have any ethical concerns with this paper?

No

Have you any concerns about statistical analyses in this paper?

Yes

Recommendation?

Accept with minor revision (please list in comments)

Comments to the Author(s)

See attached file (Appendix B).

Review form: Reviewer 3

Is the manuscript scientifically sound in its present form?

Yes

Are the interpretations and conclusions justified by the results?

Yes

Is the language acceptable?

Yes

Do you have any ethical concerns with this paper?

No

Have you any concerns about statistical analyses in this paper?

No

Recommendation?

Accept with minor revision (please list in comments)

Comments to the Author(s)

In this manuscript the authors consider the problem of resistance to the drug Temozolomide in patients with glioblastoma. They consider a multi-scale model and show that non-genetic mechanisms are sufficient to explain the evolution of resistance. The model is also parametrised with experimental data. I find this manuscript sound and interesting, and a part from a couple of minor concerns I think it is suitable for publication.

Minor points:

1. Line 45: Temozolomide is unnecessarily spelled out.
2. On line 72 it should say that "the growth rate of the cell can be selected for and lead *to* its distribution being upregulated"
3. Line 140: 's' is surely not the ability, but corresponds to the rate at which inhibition occurs. Please rephrase this.
4. Line 197-199: What is the rationale for choosing η_1 and η_2 from normal distributions? Please motivate this choice. Also the respective means (0 and 0.5) are not motivated. Please do so. Also, it is not explained what happens if η_2 becomes smaller than zero or larger than unity (which occurs with non-zero probability). Please explain this.
5. What happens to the mRNA, proteins and drug molecules at cell division? Are they split equally between the two cells? If the numbers are small then a stochastic split between the cells makes more sense. Please clarify and elaborate on this.
6. In eq. (15) what does σ and λ correspond to?
7. When working with the experimental data it is not clear how the birth and death rates are estimated. It seems as if the birth rate is estimated from growth curves, but how can the authors be certain that there is no background death rate? To my knowledge it is only possible to estimate the difference in birth and death rate from time series data. Please explain this in more detail.
8. Figure 3 is very difficult to read. The arrows are too small and it is unclear why the curves start at different points on the x-axis. Also the x-axis needs a proper label. What is given on a logarithmic scale?
9. Line 420: An alternative model with regulation is here mentioned for the first time. Why is this so?
10. Line 437: "a new parameter related to MGMT protein s ". New in what sense?
11. I suggest that the tables with the most likely parameter values are moved to the main text. Also, I suggest that the authors add confidence intervals for the values since this gives an idea of the accuracy of the fit.
12. According to eq. (16) the amount of noise increases as the rate λ decreases. Despite this, the authors suggest that inhibiting MGMT mRNA production should decrease the noise. This seems contradictory. Please explain.

Decision letter (RSOS-191243.R0)

27-Jan-2020

Dear Mr Lasri Doukkali,

The editors assigned to your paper ("Phenotypic selection through cell death: stochastic modelling of O-6-methylguanine-DNA methyltransferase dynamics") have now received

comments from reviewers. We would like you to revise your paper in accordance with the referee and Associate Editor suggestions which can be found below (not including confidential reports to the Editor). Please note this decision does not guarantee eventual acceptance.

Please submit a copy of your revised paper before 19-Feb-2020. Please note that the revision deadline will expire at 00.00am on this date. If we do not hear from you within this time then it will be assumed that the paper has been withdrawn. In exceptional circumstances, extensions may be possible if agreed with the Editorial Office in advance. We do not allow multiple rounds of revision so we urge you to make every effort to fully address all of the comments at this stage. If deemed necessary by the Editors, your manuscript will be sent back to one or more of the original reviewers for assessment. If the original reviewers are not available, we may invite new reviewers.

- Data accessibility

<http://datadryad.org/submit?journalID=RSOS&manu=RSOS-191243>

- Competing interests

- Authors' contributions

All submissions, other than those with a single author, must include an Authors' Contributions section which individually lists the specific contribution of each author. The list of Authors

should meet all of the following criteria; 1) substantial contributions to conception and design, or acquisition of data, or analysis and interpretation of data; 2) drafting the article or revising it critically for important intellectual content; and 3) final approval of the version to be published.

- Acknowledgements

- Funding statement

on behalf of Prof Mark Chaplain (Subject Editor)
openscience@royalsociety.org

Comments to Author:

Reviewers' Comments to Author:

Reviewer: 1

Comments to the Author(s)

As a general comment, if the idea underlying the modelling approach is very interesting, the manuscript lacks clarity and precision, both in regards of scientific aspects and accessibility to the reader. Therefore, I recommend a major revision and I strongly suggest you rewrite the manuscript to make it suitable for review (asking for an external reading before submitting the manuscript would be a good idea). Please find in attached file all my detailed comments.

Reviewer: 2

Comments to the Author(s)

See attached file.

Reviewer: 3

Comments to the Author(s)

In this manuscript the authors consider the problem of resistance to the drug Temozolomide in patients with glioblastoma. They consider a multi-scale model and show that non-genetic

mechanisms are sufficient to explain the evolution of resistance. The model is also parametrised with experimental data. I find this manuscript sound and interesting, and a part from a couple of minor concerns I think it is suitable for publication.

Minor points:

1. Line 45: Temozolomide is unnecessarily spelled out.
2. On line 72 it should say that “the growth rate of the cell can be selected for and lead *to* its distribution being upregulated”
3. Line 140: ‘s’ is surely not the ability, but corresponds to the rate at which inhibition occurs. Please rephrase this.
4. Line 197-199: What is the rationale for choosing η_1 and η_2 from normal distributions? Please motivate this choice. Also the respective means (0 and 0.5) are not motivated. Please do so. Also, it is not explained what happens if η_2 becomes smaller than zero or larger than unity (which occurs with non-zero probability). Please explain this.
5. What happens to the mRNA, proteins and drug molecules at cell division? Are they split equally between the two cells? If the numbers are small then a stochastic split between the cells makes more sense. Please clarify and elaborate on this.
6. In eq. (15) what does σ and λ correspond to?
7. When working with the experimental data it is not clear how the birth and death rates are estimated. It seems as if the birth rate is estimated from growth curves, but how can the authors be certain that there is no background death rate? To my knowledge it is only possible to estimate the difference in birth and death rate from time series data. Please explain this in more detail.
8. Figure 3 is very difficult to read. The arrows are too small and it is unclear why the curves start at different points on the x-axis. Also the x-axis needs a proper label. What is given on a logarithmic scale?
9. Line 420: An alternative model with regulation is here mentioned for the first time. Why is this so?
10. Line 437: “a new parameter related to MGMT protein s”. New in what sense?
11. I suggest that the tables with the most likely parameter values are moved to the main text. Also, I suggest that the authors add confidence intervals for the values since this gives an idea of the accuracy of the fit.
12. According to eq. (16) the amount of noise increases as the rate λ decreases. Despite this, the authors suggest that inhibiting MGMT mRNA production should decrease the noise. This seems contradictory. Please explain.

Author's Response to Decision Letter for (RSOS-191243.R0)

See Appendix C.

RSOS-191243.R1 (Revision)

Review form: Reviewer 1

Is the manuscript scientifically sound in its present form?

No

Are the interpretations and conclusions justified by the results?

Yes

Is the language acceptable?

Yes

Do you have any ethical concerns with this paper?

No

Have you any concerns about statistical analyses in this paper?

No

Recommendation?

Major revision is needed (please make suggestions in comments)

Comments to the Author(s)

The new version of the paper show improvement and underline better the interesting ideas carried by this work. I thank the authors for their responses to comments, which helped my understanding, and led to some useful precisions and interesting additions in the paper. However, in my opinion, further discussions and precisions are needed to reach the quality requested by RSOS.

Please find in attached file (Appendix D) some further comments.

Decision letter (RSOS-191243.R1)

15-Apr-2020

Dear Mr Lasri Doukkali:

Manuscript ID RSOS-191243.R1 entitled "Phenotypic selection through cell death: stochastic modelling of O-6-methylguanine-DNA methyltransferase dynamics" which you submitted to Royal Society Open Science, has been reviewed. The comments of the reviewer(s) are included at the bottom of this letter.

Please submit a copy of your revised paper before 08-May-2020. Please note that the revision deadline will expire at 00.00am on this date. If we do not hear from you within this time then it will be assumed that the paper has been withdrawn. In exceptional circumstances, extensions may be possible if agreed with the Editorial Office in advance. We do not allow multiple rounds of revision so we urge you to make every effort to fully address all of the comments at this stage. If deemed necessary by the Editors, your manuscript will be sent back to one or more of the original reviewers for assessment. If the original reviewers are not available we may invite new reviewers.

- Ethics statement

- Data accessibility

- Competing interests

- Authors' contributions

- Acknowledgements

- Funding statement

Kind regards,
Andrew Dunn
Royal Society Open Science Editorial Office

on behalf of Prof Mark Chaplain (Subject Editor)
openscience@royalsociety.org

Associate Editor Comments to Author:

Thank you for the efforts you have made to improve this manuscript. The Editors would like to offer a final opportunity to revise the manuscript to a publishable standard - please be aware that no further chance to revise will be offered, so please do ensure you fully address the remaining concerns of the referee. Good luck.

Reviewer comments to Author:

Reviewer: 1

Comments to the Author(s)

The new version of the paper show improvement and underline better the interesting ideas carried by this work. I thank the authors for their responses to comments, which helped my understanding, and led to some useful precisions and interesting additions in the paper.

However, in my opinion, further discussions and precisions are needed to reach the quality requested by RSOS.

Please find in attached file some further comments.

Author's Response to Decision Letter for (RSOS-191243.R1)

See Appendix E.

RSOS-191243.R2 (Revision)

Review form: Reviewer 1

Is the manuscript scientifically sound in its present form?

Yes

Are the interpretations and conclusions justified by the results?

Yes

Is the language acceptable?

Yes

Do you have any ethical concerns with this paper?

No

Have you any concerns about statistical analyses in this paper?

No

Recommendation?

Accept with minor revision (please list in comments)

Comments to the Author(s)

I thank the authors for their precisions and answers, in particular regarding experimental values used in the cell volume dynamics. I believe this study is of great interest for the readers of RSOS. Please find below some precisions that are required in my opinion to reach publication.

- About the effect of TMZ on the volume growth.
In all the paper, you refer to Eq (4), which corresponds to a TMZ-free volume growth. However, in Algo 1, step 7, you explicitly use Eq (1)-(3).
I believe you use Eq (4) only in the absence of TMZ, and use a time discretization of (1)-(3) as soon as there is TMZ. Please clarify this point to the reader, and refer to both types of Equations (not only Eq (4)).
- Eq (22): as already noted previously, the signs should be inverted. Written as it is, in the absence of TMZ the concentration in damaged DNA increases exponentially over time.
- Please define TMZ in Eqs (22) to (26). It is later defined as the "used TMZ concentration", which is not clear with respect to [TMZ] (the concentration inside the cell).
- Algo 2: you state that you use the reduced model (10),(12)-(14),(17),(18), (27) with the general error (1 and 2), while you use the full model (8)-(18) with the data-constraining error (1, 2 and 3).

In your previous answer about Epsilon, you stated "Epsilon is not a parameter of the general model, but only the reduced one (the data constrained one)".

Table 1 shows that Epsilon is not involved in the reduced model. It is involved in the data-constrained model, which is the full model according to Algo 2.

All further figures and supp figures refer to the same Equations (the reduced model), even if the models used are not the same. This is confusing to the reader. Please clarify in the paper which model you use in each case. Please also refer to the uptake function you consider in each case, since it also changes.

- Please harmonize notations between mu (supp fig) and mu_0.
- Line 303: I believe the recovery is not only slower but also smaller.
- Line 313: is it the TMZ-mediated cell death rate (kd) ?
- Line 485: parameter (typo).
- L.515: "general" model ?

Decision letter (RSOS-191243.R2)

Dear Mr Lasri Doukkali:

On behalf of the Editors, I am pleased to inform you that your Manuscript RSOS-191243.R2 entitled "Phenotypic selection through cell death: stochastic modelling of O-6-methylguanine-DNA methyltransferase dynamics" has been accepted for publication in Royal Society Open

Science subject to minor revision in accordance with the referee suggestions. Please find the referees' comments at the end of this email.

The reviewers and Subject Editor have recommended publication, but also suggest some minor revisions to your manuscript. Therefore, I invite you to respond to the comments and revise your manuscript.

- Ethics statement

- Data accessibility

If you wish to submit your supporting data or code to Dryad (<http://datadryad.org/>), or modify your current submission to dryad, please use the following link:
<http://datadryad.org/submit?journalID=RSOS&manu=RSOS-191243.R2>

- Competing interests

- Authors' contributions

- Acknowledgements

- Funding statement

Because the schedule for publication is very tight, it is a condition of publication that you submit the revised version of your manuscript before 14-Jun-2020. Please note that the revision deadline will expire at 00.00am on this date. If you do not think you will be able to meet this date please let me know immediately.

on behalf of Prof Mark Chaplain (Subject Editor)
 openscience@royalsociety.org

Associate Editor Comments to Author:

Please take care with your revision to address the remaining concerns presented by the referee.

Reviewer comments to Author:

Reviewer: 1

Comments to the Author(s)

I thank the authors for their precisions and answers, in particular regarding experimental values used in the cell volume dynamics. I believe this study is of great interest for the readers of RSOS. Please find below some precisions that are required in my opinion to reach publication.

- About the effect of TMZ on the volume growth.

In all the paper, you refer to Eq (4), which corresponds to a TMZ-free volume growth. However, in Algo 1, step 7, you explicitly use Eq (1)-(3).

I believe you use Eq (4) only in the absence of TMZ, and use a time discretization of (1)-(3) as soon as there is TMZ. Please clarify this point to the reader, and refer to both types of Equations (not only Eq (4)).

- Eq (22): as already noted previously, the signs should be inverted. Written as it is, in the absence of TMZ the concentration in damaged DNA increases exponentially over time.

- Please define TMZ in Eqs (22) to (26). It is later defined as the “used TMZ concentration”, which is not clear with respect to [TMZ] (the concentration inside the cell).

- Algo 2: you state that you use the reduced model (10),(12)-(14),(17),(18), (27) with the general error (1 and 2), while you use the full model (8)-(18) with the data-constraining error (1, 2 and 3).

In your previous answer about Epsilon, you stated “Epsilon is not a parameter of the general model, but only the reduced one (the data constrained one)”.

Table 1 shows that Epsilon is not involved in the reduced model. It is involved in the data-constrained model, which is the full model according to Algo 2.

All further figures and supp figures refer to the same Equations (the reduced model), even if the models used are not the same. This is confusing to the reader. Please clarify in the paper which model you use in each case. Please also refer to the uptake function you consider in each case, since it also changes.

- Please harmonize notations between mu (supp fig) and mu_0.

- Line 303: I believe the recovery is not only slower but also smaller.

- Line 313: is it the TMZ-mediated cell death rate (kd) ?

- Line 485: parameter (typo).

- L.515: “general” model ?

Author's Response to Decision Letter for (RSOS-191243.R2)

See Appendix F.

Decision letter (RSOS-191243.R3)

Dear Mr Lasri Doukkali,

It is a pleasure to accept your manuscript entitled "Phenotypic selection through cell death: stochastic modelling of O-6-methylguanine-DNA methyltransferase dynamics" in its current form for publication in Royal Society Open Science.

Kind regards,
Lianne Parkhouse
Editorial Coordinator
Royal Society Open Science
openscience@royalsociety.org

on behalf of the Associate Editor and Professor Mark Chaplain (Subject Editor)
openscience@royalsociety.org

Appendix A

Review of manuscript ID RSOS-191243 - Phenotypic selection through cell death: stochastic modelling of O-6-methylguanine-DNA methyltransferase dynamics

3 décembre 2019

This manuscript deals with the mathematical modelling of the dynamics of MGMT, a DNA-repair protein that is supposed to be involved in resistance to chemotherapy based on temozolomide (TMZ) for glioblastoma tumours. MGMT expression is downregulated by methylation of one of the promoters of the corresponding gene, and TMZ is thought to induce the methylation, thereby leading to cell resistance.

In this paper, the authors investigate whether a simpler explanation for TMZ-induced resistance can be found. The question of interest is: can the stochastic fluctuations associated with gene expression be enough to induce a phenotypic selection of MGMT-rich cells, that are more likely to resist to treatment ? They choose a minimal modelling approach to describe both the intracellular molecular dynamics and the cell population dynamics.

I think the idea underlying the model is interesting, but as such the manuscript does not allow for its understanding and the evaluation of its scientific soundness properly. The manuscript contains a lot of inaccuracies, and it is not entirely clear to me what the authors do in terms of modelling, numerical simulations and parameter estimation.

As a consequence, the manuscript as such clearly does not meet the requirements for publication in RSOS, and I therefore suggest to rewrite the manuscript using a more rigorous approach before submitting it again. Please find below more detailed comments.

Abstract

- It is mentioned that the "acquired phenotypic resistance were passed to daughter cells in a stable manner". This fact was not really discussed in the body of the manuscript.
- It is not clear there how the paper deals with phenotypic selection "through cell death".
- Please mention the biological experiments and link made with the model.

Background

- Explain the notion of phenotypic selection and opposition with views of genetic (this could be inspired from what is written on the subject in the Conclusion).
- If I understood properly the idea, please explain more the opposition made between resistance from MGMT-promoter methylation and resistance from death of cells having less MGMT proteins, the variability coming from stochastic fluctuations in gene expression only.

- When giving typical TMZ dosage given to patients, please provide equivalent quantities in the units used in the following.
- The biological phenomena (effect of TMZ, methylation etc) should be outlined properly.
- The part concerning literature mentions interesting existing results regarding MGMT protein levels, sensitivity to TMZ, and methylation status of the promoter of the gene. The introduction of your work and results should come after this part, and give its positioning with respect to the literature. Moreover, a comparative discussion should be added in the Conclusion part to relate to the literature.
- Paragraph starting at line 85: talking about "necessary parameters relationships for drug resistance" seems both unclear (relationship is not precise) and strong (are the parameter characteristics obtained necessary indeed, or is it simply one possible configuration ?).

Methods - introduction of the model

I could not understand properly the model, what are the sources of variability, what is done numerically. This part requires important rewriting to make it rigorous and ensure reproducibility of the model.

- Example of a clearer structure: introduce first the deterministic dynamics (chemical reactions ; cell growth & death with effect of TMZ), then the sources of stochasticity. Then, detail the algorithmic method that really makes clear how the different sources of stochasticity are treated (division, partitioning, parameters).
- Refer to the complete model in SuppMat from the beginning.
- Please provide a discussion of all biological assumptions made in a dedicated paragraph. For now I found different information in different parts of the manuscript and SuppMat.

Chemical reactions

- Page 4, line 122: you state that the MGMT is known to be stable, so that you do not account for its degradation. This is a very strong assumption, that is not properly justified: the reference you give indicates a half-life of 24h, while your experiments last at least 72h (in the SuppMat you give another reference for 60h, that does not convince to neglect protein degradation either since the order of values are the same). I suggest that you either take it into account in the reduced model (since apparently it has no real effect in the following), or discuss with stronger arguments its neglect.
- The control function of the reference protein is not clear, it should be introduced as a tool to measure phenotypic selection and why it is so. The reader understands its usefulness only at the end of page 8 with the definition of errors.

Supplementary material S1, S2

- Why is the damaged DNA still transcribed into RNA and translated into proteins ?
- Is it relevant to keep a constant proportion of DNA damage in all cells ?
- In the steady state approximation, you also assume (it should be explicit) that the repairing action of MGMT is very fast compared with the protein expression dynamics. Is it relevant ?
- Considering (S15) means that you non-dimensionalize concentrations. When considering $kd_2=1$ (line 5), you non-dimensionalize with respect to time. Therefore the range of values of parameters have a different meaning. Maybe it would be interesting to perform the non-dimensionalization of the entire problem.

- Page S2: notation star not understood.
- (S16): lacks a minus sign.
- Fig S2: to what figure do you compare it with ? I don't see how the cell viability recovery is enhanced in Fig S2.

TMZ effect on cell growth and death

- Please recall the modelling assumptions made on the effect of TMZ on the cells, and the reference towards the more complete model in SuppMat.
- The model reduction is made on strong assumptions, such as the steady state of the DNA damage dynamics. I think a paragraph explaining the reduction assumptions together with a justification would be relevant. Is it biologically relevant that over different cell cycles the proportion of damaged DNA remains constant ?
- I did not understand why the TMZ acts on the cell volume growth, and not on the division rate. Is it justified ? I found absolutely no clue of how the cell growth in volume as an effect on the dynamics of the population. It is not said what is the division rate.
- Page 4, line 135: it is not a "cell growth inhibition rate", is it a "cell volume growth rate", where TMZ has an inhibitory effect.
- It seems that later in the paper, you consider different types of models. This should be explained and clearly stated somewhere.
- Equation (9): What is T ? Is it varying in time ? Why do we speak of "slow diffusion" if there is no time derivative ? I think you consider two regimes for TMZ, but it is explicated nowhere.

Stochastic gene expression - model

- What are all the sources of stochasticity ? Are they all necessary ? Did you try to remove some of them ? A paragraph of discussion could be added.
- The numerical treatment of the model is detailed before having introduced the whole model.
- I did not understand how cell division occurs, nor molecular partitioning, while it is extremely important to understand what the MGMT proteins become.

Stochastic gene expression -algorithmic method

- The introduction of algorithms is not clear. I did not understand why the SSA is introduced, then the Tau-leaping, before stating that you choose the Gillespie algorithm.
- The methods used in these algorithms should be at least roughly explained, since you then use the notion of posterior distribution (Page 6, line 171), which cannot be understood as it is.
- If you tried different time steps, the range of values should be given (for reproducibility).
- The algorithm is not clear either. What does it do ? The quantities are not known (what is a propensity vector ? What is Tau, how is it distributed, how do you simulate it ? How does the propensity vector change ?).
- In the algorithm itself, line 3 could be after line 9, the cell volume is an input but is not used. Temp2 is not initialized.

Modelling population growth

- Is there a justification for an exponential cell volume growth ?
- Line 185: why is it constant ? Did you compare with the obtained experimental results ?
- The values used for cell volume are nonsense, since they come from measures of in vivo tumour growth during more than 2 weeks.
- What is the division rate ? Is it relevant to use a binomial process ?
- How is the volume important ? There is variability in volumes of daughter cells too, but if the reader cannot understand why is it relevant, it makes no sense.
- Line 196: what are "linear function parameters" ?
- Line 199: why do you want/need a constant population size ? Do you introduce an additional random cell death that is not taken into account in the death rate ?
- Lines 202-204: I did not understand what the tracking is about. The "intrinsic" noise is not defined either.
- Line 206: what does it imply to inherit one copy of the genes ? It seems to have no relevance for the model. Do we assume that the proportion of damaged DNA is conserved ? Is it justified ?

Parameter estimation

- Please give a reference for ABC (for example Sunnåker et al (13)).
- The method is not clear.
- Line 220 you mention "experimentally inspired prior distribution", while it is not really what you do (you choose some range of values but take a uniform distribution). Please reformulate to say that the method allows experimentally inspired prior distribution, while you simply choose uniform distributions for realistic ranges of values.
- Sentence Line 221 to 223: I did not understand this fact nor the link with the given reference.
- I did not understand Error2 because it was not clear whether the population of cell has constant size in the simulations. How do you compute the fraction of living cells ? Does it mean that dead cells are counted along the simulation ?
- At this point, it was not clear at all that you had two models, and that you wanted to consider two types of errors and why. This should be clearly said from the start. Also, concerning calibration on the data, you should make clear that you use data at two doses and test the result on the other dose values. Did you try to calibrate using for example the two lowest doses and test on the others ? In your results, you use the dose of TMZ at $800 \mu M$, while it was the state used to choose your parameters. Therefore the result has less value.
- Page 9, Line 224: it is not clear what the model output is, and what are the experimental data used to calibrate. What are the data used ?
- Why did you not consider only calibration with respect to experimental data ?
- There should be a discussion on the number of parameters to estimate and the amount of data available. Maybe a more rigorous discussion on parameter identifiability and/or sensitivity could be added. Is there a risk of overfitting ?
- What are the particles mentioned in the algorithm ?

- The precise multi-scale model should be referred to. The summary statistics is not defined. I did not understand what is θ . Explain notations.
- Does the algorithm provide only a univariate analysis, or does it select a vector of parameters ?
- Please discuss the fact that the parameters obtained in the two configurations are different. What does it mean ?

Results

- I understood that you have experimental data with population growth curves, and a model where the number of cells are constant. Is it so ? This fact should be discussed.
- Lines 241 to 244: please reformulate, since the link with the referenced experimental result is not clear.
- The ABC algorithm allows to obtain distribution of accepted parameters, which is better than having only the best fit. In my opinion it would be more interesting to first discuss statistical properties of the parameters (distribution shape, correlations) and discuss their meaning, before showing results. I would be interesting instead of choosing the best fit parameter
- Page 10, Line 250: the ratio is not defined the same as what is written in the figures. mean of the ratio is different from the ratio of the means. also in Fig6.
- Presentation of results should include a rigorous presentation of the results, a discussion of the model's behaviour and interpretation of the results.
- Line 263 (sentence about transient dose): Figure 2-A does not support the fact that under a transient TMZ dose, subsequent generation of cells can be protected by MGMT.
- Line 267: there is no steady state visible here.
- Since the dynamics is stochastic, it is relevant to perform several simulations with same parameters to show variability. It could be interesting also to have distribution of the quantities of interest at a given time to compare with data.

Statistical properties of phenotypic selection

- All the correlations are not discussed, in particular negative correlation with low p-value are important too.
- Presentation of method, observations on simulations, interpretation and biological discussion are all intertwined.
- I did not understand the use of Poisson and Birth-death processes.
- Line 304: what are the steady state mean field approximations of the model ??
- I don't think three realizations of the experiment are enough from a statistical viewpoint.
- Figure 3: please use a precise x axis. If the simulations are obtained using multiple realisations, we could have a mean trajectory together with standard errors.

Model calibration and validation using experimental data

- It was not clear before that you first used an instant infusion of TMZ, nor that you did not use the basal cell death rate either. It should be explicit and discussed.

- Fig 4: why is the curve in panel A not smoothed ? What is this curve and how did you obtain it ? At what times are displayed the histograms?
- Line 384: is the model is calibrated for TMZ=0 and 800, you cannot say that it predicts anything at this doses.
- Did you try to calibrate using other dose values before testing on others ?

Discussion and conclusion

- Please do not introduce new ideas and Figures in the conclusion, and treat them in the results part.
- The Conclusions contains a summary of modelling ideas and model construction that were not clear at all, if even present, in the body of the article. They should be given in the introduction.
- Line 422: the p-value seems large. The fact presented here is not detailed in the body of the manuscript.
- fig 7: Why negative rates ? x axis panel A is a rate.
- Please provide a discussion about the qualities and limits of the model, and how this work is positioned with respect to the litterature introduced in introduction.
- Degradation of MGMT and suicide role: did you estimate again parameters ?
- Why is your perspectives interesting ? Why do we need more microscopic descriptions ?

Appendix B

Dear Editors and Authors,

I carefully read the manuscript by the Sturrock group.

I think that this work is excellent, with a very important mix of theory and validation of theory though real data. As such, it needs only minor revisions.

Please, find below, a series of points that must be changed/included/improved:

- [1] The multiscale model ought to be better described with more systems biology important details in the full text, minimizing the supplemental materials, which describe important systems biology parts of the submitted work
- [2] In particular, concerning point 1, I do not understand whether or not the ON/OFF dynamics of gene has been included in the stochastic intracellular model. Such dynamics is important it can impact on the PD of some antitumor drugs (see Puszynski et al, PLoS Comp Biol, 2014)
- [3] Also some figures of the supplemental materials are very interesting and ought to be moved in the full text.
- [4] Model (14) ought to be better written, i.e. authors ought to write V_{i_1} and V_{i_2} instead of using twice the same symbol, and they ought to write the formula for the two offspring cells as a separate formula, not embedded in the text, as in the current version
- [5] An important detail: I suppose that author did not really use Gaussian distribution for η_1 and η_2 because this could generate negative volumes for V_F and for the volumes of the offspring cells. Not only, Gaussian distribution could generate excessively large volumes, for example $V_{i_1}=0.99 V_F$ and, as a consequence, $V_{i_1}=0.01 V_F$. I am pretty sure that they discarded values of the two stochastic processes such that the resulting volumes are negative (which in itself means that, wisely, they did not really use the true Gaussian distribution). However, are they sure that they also discarded values of the Random Variables such that the volume are too large ? More in general, it has been largely stressed in recent biophysics and systems biology literature (d'Onofrio, Alberto. "Noisy Oncology": Some Caveats in using Gaussian Noise in Mathematical Models of Chemotherapy." *Aspects of Mathematical Modelling*. Birkhäuser Basel, 2008. 229-234; d'Onofrio, Alberto, ed. *Bounded noises in physics, biology, and engineering*. Springer New York, 2013.) in mathematical oncology, and in other fields of computational biology, one ought only to use bounded noises.
- [6] The models only include at intracellular scale, intrinsic stochasticity. Extrinsic stochasticity, however, could affect the dynamics of the system (see e.g. Caravagna et al, PLoS ONE, 2013; de Franciscis et al, Scientific Report, 2016; d'Onofrio et al, Physica A, 2018) and this ought to be mentioned in the concluding remarks.
- [7] Extrinsic stochasticity has also a role in the doubly stochastic model (14), in other words η_1 and η_2 embed both intracellular stochastic phenomena and also the stochastic influence of extracellular signals. This ought to be briefly said when model (14) is proposed.
- [8] Why authors of ref 7 are written in capital letters? Please verify all bibliographic items in case there are other "anomalies"

Kind Regards,

A Referee

Appendix C

Response to referees' comments for the paper: "Phenotypic selection through cell death: stochastic modelling of O-6-methylguanine-DNA methyltransferase dynamics"

Lasri Doukkali, Ayoub; Juric, Viktorija; Verreault, Maité; Bielle, Franck; Idbaih, Ahmed; Kel, Alexander; Murphy, Brona; Sturrock, Marc

We would like to thank all three reviewers for taking the time to carefully read the manuscript and give insightful, helpful comments.

Replies to Reviewer #1

1. Abstract

1.1. Comment

It is mentioned that the "acquired phenotypic resistance were passed to daughter cells in a stable manner". This fact was not really discussed in the body of the manuscript.

Response

We thank the reviewer for raising this point. We have now added a couple of sentences to discuss this on line 297.

1.2. Comment

It is not clear there how the paper deals with phenotypic selection "through cell death".

Response

We thank the reviewer for raising this point. Without cell death there can be no phenotypic selection, we showed this in Figure S7(A) where the cell death rate was reduced to 0. We also discussed this on line 459.

1.3. Comment

Please mention the biological experiments and link made with the model

Response

We thank the reviewer for the suggestion. We have now included a sentence that links the biological experiments to the model in the abstract.

2. Background

2.1. Comment

Explain the notion of phenotypic selection and opposition with views of genetic (this could be inspired from what is written on the subject in the Conclusion). If I understood properly the idea, please explain more the opposition made between resistance from MGMT-promoter methylation and resistance from death of cells having less MGMT proteins, the variability coming from stochastic fluctuations in gene expression only.

Response

We thank the reviewer for raising this important point. We added a sentence to further highlight the difference between genetic and phenotypic selection on line 70.

2.2. Comment

When giving typical TMZ dosage given to patients, please provide equivalent quantities in the units used in the following.

Response

We thank the reviewer for this comment. 150 μM TMZ has previously been demonstrated to correspond with the levels achieved in the serum of patients during treatment, for example see, Hammond LA, Eckardt JR, Kuhn JG, Gerson SL, Johnson T, Smith L et al. A randomized phase I and pharmacological trial of sequences of 1,3-bis(2-chloroethyl)-1-nitrosourea and temozolomide.

2.3. Comment

The biological phenomena (effect of TMZ, methylation etc) should be outlined properly.

Response

This is now further outlined on line 43. We have now included an additional reference which shows that protein expression increases in response to TMZ. We do not touch upon methylation status in this particular work but have included a reference to another model which examines the relationship between TMZ and methylation status in the discussion.

2.4. Comment

The part concerning literature mentions interesting existing results regarding MGMT protein levels, sensitivity to TMZ, and methylation status of the promoter of the gene. The introduction of your work and results should come after this part, and give its positioning with respect to the literature. Moreover, a comparative discussion should be added in the Conclusion part to relate to the literature.

Response

We thank the reviewer for raising this point. We have added a couple of sentences which state how our work relates to previous literature at the end of the introduction. We have also added a comparative discussion in the conclusion part on line 540 where we compare the results of our paper with the existing literature.

2.5. Comment

Paragraph starting at line 85: talking about "necessary parameters relationships for drug resistance" seems both unclear (relationship is not precise) and strong (are the parameter characteristics obtained necessary indeed, or is it simply one possible configuration ?).

Response

We have tempered our language to "parameter relationships associated with drug resistance".

3. Methods - introduction of the model

3.1. Comment

I could not understand properly the model, what are the sources of variability, what is done numerically. This part requires important rewriting to make it rigorous and ensure reproducibility of the model. Example of a clearer structure: introduce first the deterministic dynamics (chemical reactions ; cell growth death with effect of TMZ), then the sources of stochasticity. Then, detail the algorithmic method that really makes clear how the different sources of stochasticity are treated (division, partitioning, parameters).

Response

We have changed the format to be consistent with the reviewers suggestion and included the full model dynamics as suggested by reviewer 2 also. Only the cell growth is purely deterministic though, the chemical reactions are simulated using the stochastic simulation algorithm. To clarify this we have expanded greatly upon our pseudo-code in order that the model results be reproducible.

3.2. Comment

Refer to the complete model in SuppMat from the beginning.

Response

We thanks the reviewer for raising this point. We have removed the complete model from the supp mat and included it when we first present the model in the Methods section (this is consistent with other reviewers suggestions).

3.3. Comment

Please provide a discussion of all biological assumptions made in a dedicated paragraph. For now I found different information in different parts of the manuscript and SuppMat.

Response

We have now included the full model description which was originally in the SuppMat in the main paper.

4. Chemical reactions

4.1. Comment

Page 4, line 122: you state that the MGMT is known to be stable, so that you do not account for its degradation. This is a very strong assumption, that is not properly justified: the reference you give indicates a half-life of 24h, while your experiments last at least 72h (in the SuppMat you give another reference for 60h, that does not convince to neglect protein degradation either since the order of values are the same). I suggest that you either take it into account in the reduced model (since apparently it has no real effect in the following), or discuss with stronger arguments its neglect.

Response

The reference we gave in the main text stated “MGMT is a relatively stable protein, having a half-life of more than 24h”. We agree that protein degradation can be non-negligible. This is why we included an additional Figure in the supplementary material to explore this. We found that our results did not change significantly by including protein degradation. We have now emphasise this in the main paper on line 174.

4.2. Comment

The control function of the reference protein is not clear, it should be introduced as a tool to measure phenotypic selection and why it is so. The reader understands its usefulness only at the end of page 8 with the definition of errors.

Response

We have shifted this explanation of the reference protein from page 8 to when the reference protein is first introduced.

4.3. Comment

Supplementary material S1, S2. Why is the damaged DNA still transcribed into RNAm and translated into proteins ?

Response

The DNA damage is considered as a global DNA damage term, it is not necessarily the case that the DNA was damaged precisely at the gene encoding for MGMT mRNA. However, by making the stronger assumption that damaged DNA can no longer yield mRNA for MGMT or Reference protein did not impact our results qualitatively and we found that we could recapture our initial results exactly by adjusting the transcription rate.

4.4. Comment

Is it relevant to keep a constant proportion of DNA damage in all cells ?

Response

DNA damage in the full model is modelled as a probabilistic reaction which occurs within each cell in response to the amount of TMZ administered, so can vary in a cell population. We have added a sentence on line 184 to reflect this.

4.5. Comment

In the steady state approximation, you also assume (it should be explicit) that the repairing action of MGMT is very fast compared with the protein expression dynamics. Is it relevant ?

Response

This is an interesting point. The speed of repairing the DNA damage really depends on the amount of DNA damage and depends on the type, age, and environment of the cell of interest. As a first approximation we assumed it was fast which enabled us to reduce the computational cost of simulating our model by reducing the number of reactions simulated. However, we have now modified Figure S7B in which we begin from a parameter set that yields phenotypic selection in the fast limit and explored the effect of slowing down the DNA repair rate (s) in the full model. We found that the full model deviated only from the steady state model as the DNA repair rate became very small. We added some sentences about this point on line 481. Interestingly, we also found that our parameter sets which produced phenotypic selection required very large values for DNA repair rate (s).

4.6. Comment

Considering (S15) means that you non-dimensionalize concentrations. When considering $kd_2=1$ (line 5), you non-dimensionalize with respect to time. Therefore the range of values of parameters have a different meaning. Maybe it would be interesting to perform the non-dimensionalization of the entire problem.

Response

We thank the reviewer for the comment, we accidentally omitted the units here. We have now corrected this on line 187.

4.7. Comment

Page S2: notation star not understood.

Response

We thank the reviewer for raising this point. We have now explained the notation on line 186-187.

4.8. Comment

(S16): lacks a minus sign.

Response

We thank the reviewer for reporting this - however, we could not see a missing minus sign in equation S16. We did however notice a missing “d” in differential equation S24. Perhaps this was an issue compiling the LaTeX document to a PDF.

4.9. Comment

Fig S2: to what figure do you compare it with ? I don't see how the cell viability recovery is enhanced in Fig S2.

Response

We thank the reviewer for raising this point. We compare the Fig S2(B) to Figure 2(D), this is now mentioned in the SuppMat on line 14. We also explicitly show how this cell viability recovery can be enhanced in Figure S2(C).

5. TMZ effect on cell growth and death

5.1. Comment

Please recall the modelling assumptions made on the effect of TMZ on the cells, and the reference towards the more complete model in SuppMat.

Response

We thank the reviewer for this suggestion. We have now moved the full model into the main paper. This enables us to show the modelling assumptions more fully.

5.2. Comment

The model reduction is made on strong assumptions, such as the steady state of the DNA damage dynamics. I think a paragraph explaining the reduction assumptions together with a justification would be relevant. Is it biologically relevant that over different cell cycles the proportion of damaged DNA remains constant ?

Response

We refer the reviewer to our earlier response in which we considered the impact of relaxing our assumption regarding the steady state approximation. Even in the reduced model, the cell death rate reaction allows for a lot of variability. It is encoded in a probabilistic reaction at a rate which depends on the amount of TMZ administered and the level of MGMT with the cells and therefore it can vary across a population of cells. We checked the validity of the steady state approximation under a range of different values of the parameter s , there is now some extra discussion about this in the discussion section as well as a modification to Figure S7.

5.3. Comment

I did not understand why the TMZ acts on the cell volume growth, and not on the division rate. Is it justified ? I found absolutely no clue of how the cell growth in volume as an effect on the dynamics of the population. It is not said what is the division rate.

Response

The cell growth rate and division rate are interlinked. If the cell growth rate is faster, the cells divide faster. We made an assumption that once the cell volume approximately doubles, the cells divide.

5.4. Comment

Page 4, line 135: it is not a "cell growth inhibition rate", is it a "cell volume growth rate", where TMZ has a inhibitory effect.

Response

We thank the reviewer for this point. We have now rephrased this.

5.5. Comment

It seems that later in the paper, you consider different types of models. This should be explained and clearly stated somewhere.

Response

In line with comments from reviewers 2 and 3 we have added some sentences on lines 315-316 and 442 emphasising our study of a model with ON/OFF dynamics with appropriate references in the discussion.

5.6. Comment

Equation (9): What is T ? Is it varying in time ? Why do we speak of "slow diffusion" if there is no time derivative ? I think you consider two regimes for TMZ, but it is explicited nowhere.

Response

We have rephrased “slow diffusion” to “slow uptake” and clarified the points regarding notation that the reviewer raised on line 219.

6. Stochastic gene expression - model

6.1. Comment

What are all the sources of stochasticity ? Are they all necessary ? Did you try to remove some of them ? A paragraph of discussion could be added.

Response

We thank the reviewer for this comment. We updated the pseudo code for our stochastic simulation algorithm to give some more details about our numerical implementation. In terms of the sources of stochasticity, we have intrinsic sources due to the chemical reactions as well as division noise (partitioning of molecules into daughter cells) and final volume noise (we assume there is some variability in the cell cycle). We found that the mRNA stochasticity was vital for our results and this is reflected in section “Statistical properties of phenotypic selection”.

6.2. Comment

The numerical treatment of the model is detailed before having introduced the whole model.

Response

We thank the reviewer for this comment. We have now moved the full model to the main paper so that the whole model is introduced before the numerical treatment.

6.3. Comment

I did not understand how cell division occur, nor molecular partitioning, while it is extremely important to understand what the MGMT proteins become.

Response

We thank the reviewer for this comment. We have now explain this in the extended pseudo code for the stochastic simulation algorithm as well as on lines 142.

7. Stochastic gene expression -algorithmic method

7.1. Comment

The introduction of algorithms is not clear. I did not understand why the SSA is introduced, then the Tau-leaping, before stating that you choose the Gillespie algorithm.

Response

We used the SSA for all presented simulations but used the Tau-leaping algorithm in our ABC sampling. However, if a negative quantity was encountered for a particular parameter set, we repeated the simulation using the SSA algorithm. This allowed us to explore large regions of the parameter space efficiently. This is reflected on line 226. We have replaced the Tau-leaping algorithm pseudo-code with a reference.

7.2. Comment

The methods used in these algorithms should be at least roughly explained, since you then use the notion of posterior distribution (Page 6, line 171), which cannot be understood as it is.

Response

The notion of a posterior distribution is a fundamental statistical concept. It is not something we believe should be explained in such a paper. We instead inserted an additional reference for readers unfamiliar with these concepts.

7.3. Comment

If you tried different time steps, the range of values should be given (for reproducibility).

Response

For reproducibility we included the range of values we explored on line 208.

7.4. Comment

The algorithm is not clear either. What does it do ? The quantities are not known (what is a propensity vector ? What is Tau, how is it distributed, how do you simulate it ? How does the propensity vector change ?). In the algorithm itself, line 3 could be after line 9, the cell volume is an input but is not used. Temp2 is not initialized.

Response

We decided to replace the fixed time step Tau-leap algorithm with a reference.

8. Modelling population growth

8.1. Comment

Is there a justification for an exponential cell volume growth ?

Response

This is a good point. In order to model our available growth data, we attempted to fit both an exponential and logistic growth curve. We found the exponential growth curve to be the more probable model of the data after using a Bayesian model comparison. We have inserted this point on line 128.

8.2. Comment

Line 185: why is it consistent ? Did you compare with the obtained experimental results?

Response

Ho et al. reported in their paper that TMZ inhibit the cell growth and this consistent with equation (3). We have added a couple of sentences to clarify this point on line 133.

8.3. Comment

The values used for cell volume are nonsense, since they come from measures of in vivo tumour growth during more than 2 weeks.

Response

The values found are relevant to glioblastomas and we had no other measurements available.

8.4. Comment

What is the division rate ? Is it relevant to use a binomial process ?

Response

The binomial process is a common approximation for molecular partitioning. We have now included a reference for this.

8.5. Comment

How is the volume important ? There is variability in volumes of daughter cells too, but if the reader cannot understand why is it relevant, it makes no sense.

Response

The volume is important as it dictates when division occurs. Cell division is relevant because it can lead to cell viability recovery. Cell viability recovery is important because it can be associated with drug resistance.

8.6. Comment

Line 196: what are "linear function parameters" ?

Response

These are taken from the noisy linear map model of cell growth and division which we reference on line 139.

8.7. Comment

Line 199: why do you want/need a constant population size ? Do you introduce an additional random cell death that is not taken into account in the death rate ?

Response

Essentially it is very expensive to simulate a large growing cell population and so a common approach is to simulate cells as isolated lineages (only tracking one cell as it grows and divides). However, by making this assumption the statistics of the problem are skewed. A way around this is to model the cells as a constant cell population, this solves the problem of computational expense and keeps the statistics of the simulations accurate. We clarified this on line 153.

8.8. Comment

Lines 202-204: I did not understand what the tracking is about. The "intrinsic" noise is not defined either.

Response

We changed the word tracking to "simulating" to avoid any confusion. We added a brief definition for intrinsic noise after the term is first used also.

8.9. Comment

Line 206: what does it imply to inherit one copy of the genes ? It seems to have no relevance for the model. Do we assume that the proportion of damaged DNA is conserved ? Is it justified ?

Response

For simplicity, we have assumed that each daughter cell inherits one copy of the gene from the mother cell upon division and no mutation occurs during cell division. We made this explicit on line 159 to rule out any genetic mutation events. The proportion of cells with damaged DNA varies in the cell population.

9. Parameter estimation

9.1. Comment

Please give a reference for ABC (for example Sunnâker et al (13)).

Response

We thank the reviewer for this suggestion. the reference is now included on line 225.

9.2. Comment

The method is not clear.

Response

We have now endeavoured to clarify the method.

9.3. Comment

Line 220 you mention "experimentally inspired prior distribution", while it is not really what you do (you choose some range of values but take a uniform distribution). Please reformulate to say that the method allows experimentally inspired prior distribution, while you simply choose uniform distributions for realistic ranges of values.

Response

We agree with the reviewer and have adjusted our language accordingly.

9.4. *Comment*

Sentence Line 221 to 223: I did not understand this fact nor the link with the given reference.

Response

We have clarified this and explained the link with the reference on line 226.

9.5. *Comment*

I did not understand Error2 because it was not clear whether the population of cell has constant size in the simulations. How do you compute the fraction of living cells ? Does it mean that dead cells are counted along the simulation ?

Response

We thank the reviewer for this comment - we have now written some additional sentences explaining that we simulate a cell population of maximum size N on line 154. However, if there are cell death events then this can drop below N (while subsequent cell division can then replenish the population). The fraction of living cells at time t is computed as follow: $N(t) = M(t)/K$ where $M(t)$ is the number of living cells and K is the total cell population simulated. We have inserted this clarification before introducing Error2 on line 161.

9.6. *Comment*

At this point, it was not clear at all that you had two models, and that you wanted to consider two types of errors and why. This should be clearly said from the start. Also, concerning calibration on the data, you should make clear that you use data at two doses and test the result on the other dose values. Did you try to calibrate using for example the two lowest doses and test on the others ? In your results, you use the dose of TMZ at 800 M, while it was the state used to choose your parameters. Therefore the result has less value.

Response

We have inserted some justification for our two different ABC fittings, one without experimental data and one with experimental data. Regarding the former, we wanted to uncover theoretical relationships between parameters that were necessary to give qualitative behaviour consistent with emergent gene expression. This general ABC fitting yielded parameter distributions that were illuminating for understanding the model behaviour. In particular, this allowed us to gain insight into how noisy mRNA expression was needed for example. Regarding the latter, we could have calibrated our model with experimental data in a number of different ways but we had to choose one. We sought to use some subset of the data for calibration and save some for validation. We presented some results for the 800 μM case but we could have shown results for intermediary values also, which as we showed in Figure 6 also gave good fits.

9.7. *Comment*

Page 9, Line 224: it is not clear what the model output is, and what are the experimental data used to calibrate. What are the data used ?

Response

We have clarified what the model output is on page 9, Line 243. What we meant by model output cell viability time series was the fraction of living cells for TMZ = 0 and 800 μM . The data used was from the cell viability assay result (shown in Figure 4(B)) for two TMZ concentrations 0 and 800 μM .

9.8. *Comment*

Why did you not consider only calibration with respect to experimental data ?

Response

We addressed this point previously. We wanted to first learn about general parameter relationships necessary for phenotypic selection.

9.9. Comment

There should be a discussion on the number of parameters to estimate and the amount of data available. Maybe a more rigorous discussion on parameter identifiability and/or sensitivity could be added. Is there a risk of overfitting ?.

Response

We have expanded our discussion to include a sentence about what we believe would help constrain our model parameters more.

9.10. Comment

What are the particles mentioned in the algorithm ?

Response

The particles refer to individual parameter sets. We have clarified this on line 250.

9.11. Comment

The precise multi-scale model should be referred to. The summary statistics is not defined. I did not understand what is θ . Explain notations.

Response

The summary statistics in this case is the Error. We introduced a couple of sentences to clarify this and explain notations on line 250.

9.12. Comment

Does the algorithm provide only a univariate analysis, or does it select a vector of parameters ?

Response

A vector of parameters is explored - all parameters are randomly sampled simultaneously. A beneficial aspect of ABC is that we can perform a global sensitivity analysis while simultaneously fitting the model to data.

9.13. Comment

Please discuss the fact that the parameters obtained in the two configurations are different. What does it mean ?

Response

We would not expect these two different parameter studies to have similar results and it is not a fair comparison to compare them as they have different numbers of parameters and use different error functions. Interestingly we did find that they both favoured a small transcription rate (which we commented on in line 491.)

10. Results

10.1. Comment

I understood that you have experimental data with population growth curves, and a model where the number of cells are constant. Is it so ? This fact should be discussed.

Response

We thank the reviewer for raising this point. While the total number of cells we simulated had a maximum, if a cell division event occurred when the cell population was at this maximum we would randomly replace a cell in the population with the new daughter cell. This is a technique that ensures the gene expression statistics generated by our model are accurate and saves the computational cost of simulating an ever-growing population of cells. We simply used the experimental data to inform the exponential growth rate. We added a couple of sentences to discuss this on line 99.

10.2. Comment

Lines 241 to 244: please reformulate, since the link with the referenced experimental result is not clear.

Response

We thank the reviewer for raising this point. This section is moved to line 178, where we first mention REF protein, as suggested by the reviewer.

10.3. Comment

The ABC algorithm allows to obtain distribution of accepted parameters, which is better than having only the best fit. In my opinion it would be more interesting to first discuss statistical properties of the parameters (distribution shape, correlations) and discuss their meaning, before showing results. I would be interesting instead of choosing the best fit parameter

Response

We agree with the reviewer about the strength of ABC and have adjusted the text to describe the relationships between parameters in more detail on line 313.

10.4. Comment

Page 10, Line 250: the ratio is not defined the same as what is written in the figures. mean of the ratio is different from the ratio of the means. also in Fig6.

Response

We thank the reviewer for pointing this out. We have now corrected these anomalies in the y-axes labels throughout.

10.5. Comment

Presentation of results should include a rigorous presentation of the results, a discussion of the model's behaviour and interpretation of the results.

Response

We have added in some more sentences discussion the model's behaviour and interpretations of the results on lines 430-438.

10.6. Comment

Line 263 (sentence about transient dose): Figure 2-A does not support the fact that under a transient TMZ dose, subsequent generation of cells can be protected by MGMT.

Response

Figure 2(A) shows how the mean of MGMT to the REF increases in response to a transient TMZ dose. This suggests that following this TMZ dose, cells can be protected due to the elevated levels of MGMT - as the cell death rate is inversely proportional to the amount of MGMT whereas if the cells had not received a TMZ dose their level of MGMT would have remained lower and therefore faced a higher cell death rate if TMZ was administered again.

10.7. Comment

Line 267: there is no steady state visible here.

Response

We have extended this simulation so that a steady state is more clearly achieved (Fig.1).

10.8. Comment

Since the dynamics is stochastic, it is relevant to perform several simulations with same parameters to show variability. It could be interesting also to have distribution of the quantities of interest at a given time to compare with data.

Fig. 1: Dose-dependent, reversible upregulation of MGMT plot showing simulations from model defined by chemical reactions (10), (12)-(14), (17), (18) and (27) coupled with ODE (4) using parameters defined in Table 1 and initial conditions in Table S1 and TMZ uptake modelled by f_4 with $x = 144$ hours. We include this plot here to verify that a steady state is indeed reached.

Response

While what the reviewer says is true for individual cells, we are presenting population level results. As the reviewer suggests, it is interesting to produce distributions of the quantities of interest at a given time which is precisely what we have done in Figures 5A.

11. Statistical properties of phenotypic selection

11.1. Comment

All the correlations are not discussed, in particular negative correlation with low p-value are important too.

Response

We agree with the reviewer and have now included additional discussion about the correlations we did not mention.

11.2. Comment

Presentation of method, observations on simulations, interpretation and biological discussion are all intertwined. I did not understand the use of Poisson and Birth-death processes.

Response

We have attempted to clarify these points on pages 4-5-6 and 13 so that a broader readership can follow. We also created a new subsection in the Methods section called “Statistical measures” to help ease reading.

11.3. Comment

Line 304: what are the steady state mean field approximations of the model ??

Response

It is possible to replace the stochastic simulation algorithm with an analogous ODE system which represents the mean field dynamics of the stochastic system. We have attempted to clarify this on line 317 and added a reference for interested readers unfamiliar with this.

11.4. Comment

I don't think three realizations of the experiment are enough from a statistical viewpoint.

Response

We agree that three realizations would be too little to draw any statistically significant conclusions from. However, these are realisations at the population level, with each “realisation” corresponding to 10,000 cells.

11.5. Comment

Figure 3: please use a precise x axis. If the simulations are obtained using multiple realisations, we could have a mean trajectory together with standard errors.

Response

We have updated figure 3. The simulations are obtained for noise/skewness values computed across a population of 10,000 cells.

12. Model calibration and validation using experimental data

12.1. Comment

It was not clear before that you first used an instant infusion of TMZ, nor that you did not use the basal cell death rate either. It should be explicated and discussed.

Response

We have explicitly included these details on pages 10 and 11.

12.2. Comment

Fig 4: why is the curve in panel A not smoothed ? What is this curve and how did you obtain it ? At what times are displayed the histograms?

Response

The curve in Figure 4A refers to our in house experimental growth data. Since we only had data for 7 time points, it appears somewhat disjointed. We have now updated the histograms in Figure 5 to include the precise times as opposed to just “Before TMZ” and “After TMZ”.

12.3. Comment

Line 384: is the model is calibrated for TMZ=0 and 800, you cannot say that it predicts anything at this doses.

Response

That's correct. We have corrected any text that claims we are predicting behaviour for TMZ = 0 and 800.

12.4. Comment

Did you try to calibrate using other dose values before testing on others ?

Response

We did not, we chose to calibrate using only TMZ = 0 and 800. This led to a somewhat poor fit for the TMZ = 600 case which we may have been able to get closer to had we chosen a different calibration. However, we were mindful of not overfitting the model and also leaving enough data for validation.

13. Discussion and conclusion

13.1. Comment

Please do not introduce new ideas and Figures in the conclusion, and treat them in the results part.

Response

We have shifted some of the figures we introduced in the discussion to the supplemental material.

13.2. Comment

The Conclusions contains a summary of modelling ideas and model construction that were not clear at all, if even present, in the body of the article. They should be given in the introduction.

Response

We have added some text in a similar style to the conclusions to the earlier part of the manuscript.

13.3. Comment

Line 422: the p-value seems large. The fact presented here is not detailed in the body of the manuscript.

Response

We thank the reviewer for this comment. We have now clarified this section and stressed the fact that “p” doesn’t refer to a p-value, but to the model probability derived from the application of ABC model comparison.

13.4. Comment

fig 7: Why negative rates ? x axis panel A is a rate.

Response

We thank the reviewer for this comment. We have now clarified this point in the figure legend. The rate is plotted on a base 10 logarithmic scale.

13.5. Comment

Please provide a discussion about the qualities and limits of the model, and how this work is positioned with respect to the litterature introduced in introduction.

Response

We have added some more discussion regarding these suggestions.

13.6. Comment

Degradation of MGMT and suicide role: did you estimate again parameters?

Response

Due to limited computational resources we did not redo these estimates. However, we did check the impact of including suicide enzymatic activity and active protein degradation using individual parameter sets from our final posterior distribution estimates and we found the same qualitative behaviour. This is something we will investigate further in future work.

13.7. Comment

Why is your perspectives interesting ? Why do we need more microscopic descriptions ?

Response

We have added some final sentences highlighting more why our perspectives are interesting.

Replies to Reviewer #2

Comment

The multiscale model ought to be better described with more systems biology important details in the full text, minimizing the supplemental materials, which describe important systems biology parts of the submitted work

Response

We thank the reviewer for this suggestion and have endeavoured to include as much of the supplementary materials into the main text as possible.

13.8. Comment

In particular, concerning point 1, I do not understand whether or not the ON/OFF dynamics of gene has been included in the stochastic intracellular model. Such dynamics is important it can impact on the PD of some antitumor drugs (see Puszynski et al, PLoS Comp Biol, 2014)

Response

We simply used a constitutive gene expression model where the gene is always "ON". However, the ON/OFF dynamics are very interesting. We found that we could find very similar behaviour from our model by incorporating such dynamics, however when we performed a Bayesian model selection using both the constitutive gene expression model and the model with the ON/OFF dynamics, the constitutive model came out as the winner. We had a solitary sentence in the discussion explaining this but have now expanded upon this and included the reference that the reviewer suggested on line 445.

13.9. Comment

Also some figures of the supplemental materials are very interesting and ought to be moved in the full text.

Response

We agree with the reviewer and have now included some more. Specifically, we included the model derivation and parameter tables in the methods section of the main paper.

13.10. Comment

Model (14) ought to be better written, i.e. authors ought to write V_{I_1} and V_{I_2} instead of using twice the same symbol, and they ought to write the formula for the two offspring cells as a separate formula, not embedded in the text, as in the current version

Response

We agree with the reviewer and believe this to be an excellent suggestion. We have now included this more precise notation and extra formula near line 142.

13.11. Comment

*An important detail: I suppose that author did not really use Gaussian distribution for η_1 and η_2 because this could generate negative volumes for V_F and for the volumes of the offspring cells. Not only, Gaussian distribution could generate excessively large volumes, for example $V_{I_1} = 0.99V_F$ and, as a consequence, $V_{I_1} = 0.01V_F$. I am pretty sure that they discarded values of the two stochastic processes such that the resulting volumes are negative (which in itself means that, wisely, they did not really use the true Gaussian distribution). However, are they sure that they also discarded values of the Random Variables such that the volume are too large? More in general, it has been largely stressed in recent biophysics and systems biology literature (d'Onofrio, Alberto. "Noisy Oncology": Some Caveats in using Gaussian Noise in Mathematical Models of Chemotherapy." *Aspects of Mathematical Modelling*. Birkhäuser Basel, 2008. 229-234; d'Onofrio, Alberto, ed. *Bounded noises in physics, biology, and engineering*. Springer New York, 2013) in mathematical oncology, and in other fields of computational biology, one ought only to use bounded noises.*

Response

We thank the reviewer for raising this important detail. However, in all our simulations we never encountered such trouble as the standard deviation which we chose for η_1 and η_2 were sufficiently small so as to avoid these issues. We demonstrate random sampling of these distributions below as well as the final cell volume distribution. However, we also performed some simulations with bounded noises (where if a final volume generated was outside of some finite range, the distribution was resampled) and have made a note of this near line 141 along with the citation the reviewer raised.

Fig. 2: Typical values of division noise sampled for chosen parameter values in manuscript

Fig. 3: Typical values of final cell volume for parameter values in manuscript

Fig. 4: Typical values of final cell volume for bounded noise case.

13.12. Comment

The models only include at intracellular scale, intrinsic stochasticity. Extrinsic stochasticity, however, could affect the dynamics of the system (see e.g. Caravagna et al, PLoS ONE, 2013; de Franciscis et al, Scientific Report, 2016; d’Onofrio et al, Physica A, 2018) and this ought to be mentioned in the concluding remarks. Extrinsic stochasticity has also a role in the doubly stochastic model (14), in other words η_1 and η_2 embed both intracellular stochastic phenomena and also the stochastic influence of extracellular signals. This ought to be briefly said when model (14) is proposed.

Response

These are very interesting points and something we had started to explore in more detail. We have now extended our future work section to include a sentence about extrinsic noise with the citations mentioned. We also mention the embedded extrinsic stochasticity after introducing η_1 and η_2 on line 149.

13.13. Comment

Why authors of ref 7 are written in capital letters? Please verify all bibliographic items in case there are other “anomalies”

Response

We thank the reviewer for spotting this issue and have now double checked over the rest of the bibliographic items to eliminate any “anomalies”.

Replies to Reviewer #3

13.14. Comment

Line 45: Temozolomide is unnecessarily spelled out.

Response

We thank the reviewer for spotting this. This is now corrected in the manuscript.

13.15. Comment

*On line 72 it should say that “the growth rate of the cell can be selected for and lead *to* its distribution being upregulated”*

Response

We thank the reviewer for spotting this. This is now corrected in the manuscript on line 77.

13.16. Comment

Line 140: ‘s’ is surely not the ability, but corresponds to the rate at which inhibition occurs. Please rephrase this.

Response

We thank the reviewer for pointing this out. This is now rephrased in the manuscript on line 186.

13.17. Comment

Line 197-199: What is the rationale for choosing η_1 and η_2 from normal distributions? Please motivate this choice. Also the respective means (0 and 0.5) are not motivated. Please do so. Also, it is not explained what happens if η_2 becomes smaller than zero or larger than unity (which occurs with non-zero probability). Please explain this.

Response

We chose η_1 and η_2 to be consistent with papers *Tanouchi et al, Nature, 2015* and *Bertaux et al, RSOS, 2018*. Our motivation for sampling from these distributions was driven by the fact that we wanted to reflect some variability in the cell division process and believed this to be a good first approximation. The choice of mean 0 for η_1 was to allow the cell division event to occur either a little bit before or after precise volume doubling and the choice of mean 0.5 for η_2 was to ensure the cells approximately half when cell division takes place. We have now included additional lines clarifying these points in the text on line 191. The choice of η_2 is inspired by experimental literature and we include a diagram here to show the range of possible values from 10,000 samples. In line with the suggestions of reviewer 2, we also investigated the impact of simulating cell division using bounded noises. I.e, if the division noise sampled was outside the range (0.45,0.55) we would resample. We found that our results remained the same using this bounded approach.

13.18. Comment

What happens to the mRNA, proteins and drug molecules at cell division? Are they split equally between the two cells? If the numbers are small then a stochastic split between the cells makes more sense. Please clarify and elaborate on this.

Response

The mRNA and protein molecules are binomially partitioned between the two daughter cells. In order to capture asymmetric cell division, we assumed that cells were not divided exactly in two every upon cell division but instead the division ratio was sampled from a normal distribution with mean 0.5 and standard deviation 0.05. This captured some variability in this event and also informed the partitioning. The mRNA and protein molecules were partitioned between daughter cells via a binomial distribution with the same corresponding division ratio. This meant that if one daughter cell happened to be larger than the other, then it would be more likely to inherit more mRNA and protein molecules. For simplicity, we assumed that the drug was constantly applied throughout our simulations.

Fig. 5: Typical values of division noise sampled for chosen parameter values in manuscript

13.19. *Comment*

In eq. (15) what does σ and λ correspond to?

Response

σ represents the standard deviation and λ represents the mean. We have now updated the text to reflect this near line 260.

13.20. *Comment*

When working with the experimental data it is not clear how the birth and death rates are estimated. It seems as if the birth rate is estimated from growth curves, but how can the authors be certain that there is no background death rate? To my knowledge it is only possible to estimate the difference in birth and death rate from time series data. Please explain this in more detail.

Response

We thank the reviewer for raising this interesting point. We fitted both the exponential growth equation and logistic growth equation to the growth curves. The logistic growth equation implicitly accounts for background cell death. We found no difference in the overall growth rate estimate regardless of the model we fitted to the data. Therefore we felt confident in our estimate of the growth rate. We have clarified these points near line 128.

13.21. *Comment*

Figure 3 is very difficult to read. The arrows are too small and it is unclear why the curves start at different points on the x-axis. Also the x-axis needs a proper label. What is given on a logarithmic scale?

Response

We have now updated this figure in line with the reviewer's comments. The curves begin at different points in Figure 3A due to the noise and skewness being computed at different time points. The arrows in the figure indicate this. We have updated the figure legend to clarify this.

13.22. *Comment*

Line 420: An alternative model with regulation is here mentioned for the first time. Why is this so?

Response

We felt it may be nice to show additional insight into the problem and some readers are interested in this on/off switching dynamics (for example, reviewer 2). We included it at the end so as not to distract the reader with other models from the beginning.

13.23. Comment

Line 437: “a new parameter related to MGMT protein s”. New in what sense?

Response

We have now reworded this on line 475.

13.24. Comment

I suggest that the tables with the most likely parameter values are moved to the main text. Also, I suggest that the authors add confidence intervals for the values since this gives an idea of the accuracy of the fit.

Response

This is an excellent suggestion from the reviewer which we have now incorporated into the main text on page 12. We tried to add confidence intervals for each of the tables but some of the parameters were very ill-constrained leading to huge confidence intervals, instead we decided to refer the reader directly to the posterior distributions which contain more information.

13.25. Comment

According to eq. (16) the amount of noise increases as the rate lambda decreases. Despite this, the authors suggest that inhibiting MGMT mRNA production should decrease the noise. This seems contradictory. Please explain.

Response

This is well spotted. The word “inhibit” was chosen incorrectly here. Instead we wanted to suggest that a synthetic negative feedback could decrease the noise as in Singh et al. We have now updated the text on line 527 to avoid this confusion.

Appendix D

REVIEW OF MANUSCRIPT ID RSOS-191243.R1

FURTHER COMMENTS ON AUTHORS' RESPONSES

- *Supplementary material S1, S2. Why is the damaged DNA still transcribed into RNAm and translated into proteins ?*

Response

The DNA damage is considered as a global DNA damage term, it is not necessarily the case that the DNA was damaged precisely at the gene encoding for MGMT mRNA. However, by making the stronger assumption that damaged DNA can no longer yield mRNA for MGMT or Reference protein did not impact our results qualitatively and we found that we could recapture our initial results exactly by adjusting the transcription rate.

 I did not understand how relevant it was to keep these equations, since you do not comment on them, and you get rid of them in the reduced model without any discussion. Therefore, since neglecting them does not change qualitatively the results, I would suggest to remove them while justifying their not changing the results in SuppInfo (as for protein degradation). This would allow for more clarity in the model description. Please either discuss this choice, or simply consider that the damaged DNA can not be transcribed to RNAm in the full model.

- *In the steady state approximation, you also assume (it should be explicit) that the repairing action of MGMT is very fast compared with the protein expression dynamics. Is it relevant ?*

Response

This is an interesting point. The speed of repairing the DNA damage really depends on the amount of DNA damage and depends on the type, age, and environment of the cell of interest. As a first approximation we assumed it was fast which enabled us to reduce the computational cost of simulating our model by reducing the number of reactions simulated. However, we have now modified Figure S7B in which we begin from a parameter set that yields phenotypic selection in the fast limit and explored the effect of slowing down the DNA repair rate (s) in the full model.

We found that the full model deviated only from the steady state model as the DNA repair rate became very small. We added some sentences about this point on line 481. Interestingly, we also found that our parameter sets which produced phenotypic selection required very large values for DNA repair rate (s).

 Thank you for this interesting response. Can you therefore add a sentence when presenting the steady state approximation about this fact (the repairing action assumed also fast, and its relevance)?

- *About Eq (S16), now (22): when TMZ acts on DNA, it produces more DNA_Dam, so both signs should be inverted if I am not mistaken.*

- *I did not understand Error2 because it was not clear whether the population of cell has constant size in the simulations. How do you compute the fraction of living cells ? Does it mean that dead cells are counted along the simulation ?*

Response

We thank the reviewer for this comment - we have now written some additional sentences explaining that we simulate a cell population of maximum size N on line 154. However, if there are cell death events then this can drop below N (while subsequent cell division can then replenish the population). The fraction of living cells at time t is computed as follow: $N(t) = M(t)/K$ where $M(t)$ is the number of living cells and K is the total cell population simulated. We have inserted this clarification before introducing Error2 on line 161.

 Therefore, when cell divisions occur, do new cells most frequently replace dead cells ? If the ref [38] given indeed deals with exponentially growing populations, I understood that their point is to compare the analysis of an isolated lineage with the analysis of a snapshot of the population at a given time. I did not understand that they randomly eliminate some cells. Please precise your point regarding this issue, or remove the end of sentence l.155 about better accuracy.

Is it following this constant population assumption that you do not consider a basal cell death in the first "full" model ? Is it so that N is not a bit damped by basal cell death ?

I think this is interesting to the readers, and further indication about the interaction between cell division and cell deaths in the counting could be added.

- Fig 4: why is the curve in panel A not smoothed ? What is this curve and how did you obtain it ? At what times are displayed the histograms?

Response

The curve in Figure 4A refers to our in house experimental growth data. Since we only had data for 7 time points, it appears somewhat disjointed. We have now updated the histograms in Figure 5 to include the precise times as opposed to just "Before TMZ" and "After TMZ".

 I still did not understand how the curve is obtained.
The legend states that it follows from fitting an exponential curve, therefore why not plotting the fitted exponential curve ?

- Degradation of MGMT and suicide role: did you estimate again parameters?

Response

Due to limited computational resources we did not redo these estimates. However, we did check the impact of including suicide enzymatic activity and active protein degradation using individual parameter sets from our final posterior distribution estimates and we found the same qualitative behavior. This is something we will investigate further in future work.

 In my opinion this should be precised in the paper.

- Does the algorithm provide only a univariate analysis, or does it select a vector of parameters ?

Response

A vector of parameters is explored - all parameters are randomly sampled simultaneously. A beneficial aspect of ABC is that we can perform a global sensitivity analysis while simultaneously fitting the model to data.

 Therefore parameters are selected as a vector.

Is the "best fit" defined by the recollection of the modes of each distribution (therefore not taking into account that the combinations of values are selected)?

If so, do you observe some vectors θ^* that are close to this choice ?

OTHER ISSUES

A) About the cell volume growth

a) In the first paragraph of Background, please explain how you relate growth curves for the numbers of cells with a model for individual cell volume growth.

As I understand it, you fit an exponential model on the data for counts of cells.

Then you use this as a justification to model the volume growth of a single-cell with an exponential law. I do not understand how this is relevant.

Why not simply use a mean time before division to simulate a stochastic cell division process. Then model the effect of TMZ on this quantity.

For cell partitioning, using the same random sampling, one can simulate random partitioning with random bias to produce similar effects.

A typical time before division could perhaps be more realistically related to the cell culture data.

If not, please provide more explanations.

b) I.136-137: please explain and discuss how you use the volumes of in vivo tumors over 2 weeks

in relation with an equation for single-cell volume growth. Why do you need them ?

Is it related to V_I and V_F ? The text you added in the paragraph between (5) and (6) shows that you actually consider that a single-cell has a volume of about 30mL.

Why not instead consider a realistic typical volume of a cell, and assume that it divides when its volume reaches twice the initial value ?

c) However, in Algo 1, you consider $V_i(0)=1$, is it related to V_I ?

B) About referring precisely to the models and methods used.

You introduced two models, and (if I understood properly), you follow two parameter estimation procedures, that you also refer to using the word "model".

The term 'model' is also sometimes found alone, e.g in the conclusion, and it is not clear whether you talk of one of the models, or of your global modeling approach.

I found it still difficult to understand what is used, when, and why.

If your responses show your confidence and knowledge about the procedures followed, I unfortunately miss such clarity in the paper. This is a remark of form, but that is in my opinion critical for understanding and reproducibility.

As an example, following information of Algo 2, the "reduced model" is the "general" model, associated with Errors 1&2 and Table 1, while the "full" model is the "data-constrained model", that is related to Errors 1,2&3 and to Table 2.

However, this is the reduced model that is mentioned in Fig 5 related to calibration using experimental data (i.e Error 3).

-l.202 & algo 1: you seem to say that only the reduced model is simulated.

Please be very specific by using each time the same name and refer to the corresponding equations, and type of calibration or Errors used in the calibration. Make it as specific as possible, even in the Discussion part.

Also:

- the TMZ uptake function is a part of the model, and should be referred to.

- All along the draft, you refer to (4) for the cell volume growth, meaning that you do not consider the inhibition of cell growth by TMZ (introduced in (3)).

However, later on you show results related to (3) (FigS3), so that we can not understand what you really considered.

- Please structure the discussion around the main ideas that follow from you work.

C) Analysis of the results

In my opinion, the work done is very interesting, and there is a lot to discuss.

However, there is to me a lack for analysis and interpretation of the results, most likely the observations are simply stated.

For example, when discussing parameter correlations and sensitivity, some correlations are simply mentioned without further analysis (l.313), and no link is made with the sensitivity of the parameters involved, which could be interesting.

REMARKS AND QUESTIONS

- Please introduce Equation (16) for cell death, and discuss why you choose to neglect basal cell death in the full model while you consider it in the reduced model.

More generally, is it useful to use the full model ? I feel that the comparison made in SuppInfo can be enough, which would allow to perform both types of parameter estimation on the same model.

- l.187 (taking $kd_2=1$). It amounts to having a different parameter value in

the numerator of the TMZ-mediated cell death rate (instead of k_d). It could be interesting to check whether changing only this value could change a lot the dynamics.

- When computing Error1 (the ratio), you compute means over the population separately for REF and MGMT. Did you try computing $\langle \text{REF}/\text{MGMT} \rangle$ to have the ratio in each cell ? It would be interesting to see the histogram of such quantity.

- I am not very comfortable with the frequent use of the term "need" when commenting the parameter relations selected during the parameter estimation. Can we say that some relation is needed for phenotype selection, when the estimation procedure aimed at minimizing an error function carrying other constraints ?

- I am not sure to agree with your interpretation for μ and I_{50} in Fig S3. When looking at Errors 1&2, why does TMZ have to inhibit cell volume growth ? Later on you state that I_{50} is not sensitive, does it play a role here ? Also, the correlation plot can be split into two distinct parts, which can be discussed. For example, maybe when I_{50} is too large, then TMZ does not have a big effect for low concentration values, so that the MGMT are less likely to be key players. Maybe the fact that we witness some large values of I_{50} (though less frequently than lower values) is related to Error 2 and cell viability.

Manual variation of the level of mRNA noise:

- If I understood correctly, you vary the noise and skew levels by changing b_1 .

Therefore, do you take the new b_2 as equal to $\text{old}_b(b_1*b_2)/\text{new}_b1$?

If (as I understood) the noise and skew are modified together while modifying b_1 , both effects should

be investigated together, and therefore maybe a 3D plot would be clearer.

- Fig 3: it could be clearer to use points for each values investigated ?

- Did you try without any random bias in partitioning ?

This would allow focusing on the noise related to the protein activity alone.

- Did you try fixing the most sensitive parameters, and retry the first parameter estimation for the others ?

Results

- l.283-284: "MGMT returns to the initial amount". It seems only true for Fig2-A).

The selection seems reversible for A, and dose-dep and partially reversible for B.

These lines could be detailed a bit more.

- l.284 It seems like it is the numerical cell doubling time that matters here, not the one for glioma. Can you show your estimate for this value ? It would also support your comment on line 300. Maybe you could define it and explain how you compute it.

Discussion

- l.434 "All components of the model are necessary". This sentence seems a bit too strong in my opinion. What about epsilon ? The TMZ acting on cell volume growth ?

- l.526: if there is a negative feedback loop for transcription, it still leads to lower mRNA values, so more noise. It could however lead to less proteins. Can you explain a bit more how it works?

MINOR ISSUES

Background

- l.90: size -> viability.
- l.100: the cell growth rate and THE impact.
- End of background: please add a conclusion to state the outcome of the work.

Modelling cell growth and division

- Eq (3): please introduce [TMZ] and I_{50} . Is it the pool concentration directly ?
 - Also, a 't' is missing in Equation (4).
 - Please precise how the division time of a cell is obtained.
 - Eq (5): please precise if the parameters a and b are individual or population parameters.
 - After (5): please do not start a sentence with a symbol (η_1).
 - "..., we also simulated our final results...": please be more specific about "final results".
 - l.165: Following your notations, the cell viability recovery is defined with respect to the cell viability, not the number of living cells. Please correct either the definition or the words used.
- Please also define/precise how you obtain the "minimal" number following TMZ addition. Over what is it minimal ?

Modelling intracellular processes

- l.173: if you do not take them into account in the main work, please shift (19) and (20) to the SuppInfo for clarity. Nevertheless I found that the justification in SuppInfo was satisfying, and prevent any doubts on this assumption.
- l.184-185: please introduce the notion of damage by alkylation, for clarity with respect to the rest of the paper.
- After (20): is the "pseudo steady-state approximation" the same that the one you call later on "steady-state approximation"?
- p.8, please remove the capital letter at the beginning of the first line.
- After (25), "the reaction defined in (16)".
- l.209 to 217: this part seems more adapted to the Parameter Inference part.

Modeling TMZ uptake

- Please put Modeling TMZ uptake before the schematic diagram and the Algo 1, and explain the TMZ pool. What is equal to f_3 or f_4 ?
- The notations of f_3 and f_4 should show if they depend on time.
- The notation K is already used.
- The notation x is a bit misleading, perhaps t_{in} and t_{out} could be more specific ?
- The inequalities defining f_4 and f_5 are not correct wrt the text.

Algo 1:

- In 5., how are the "smallest" indexes defined ?
- In 7., in the exponential, is it τ rather than $t + \tau$? Or $V_i(t)$ should be $V_i(0)$?
- L_F is V_F ?
- In 8., F is t_f ?
- Equation (4) is not an ODE.

Parameter estimation

- l.237: "mean following TMZ administration". This is unclear, since this quantity is also computed before TMZ administration.
- l.247 to 250: it would be useful at this point to present what you do next, and maybe to clarify the two procedures you follow.

Algo 2:

1. Notation epsilon already used.
 2. Notation $M(x|\theta)$ rather than x/θ .
- Please precise the value for N_p .

Statistical measures

- This part may be more useful at the point where you introduce the variation in mRNA noise.

Table 2:

- Please clarify the range for the growth rate. Referring to Fig 4 where no range is explicit is not enough.
- How did you choose the parameters for TMZ uptake here ?

Results

- l.277: f_5 is not mentioned, though used in Figure 2, and commented in line 282. Also, if you comment the simulations using f_4 first, maybe it is better to switch columns in the Fig ?
- l.279 to 281: this seems not useful at this point. It would be clearer either before when introducing the uptake function, or later on when it is used.
- l.296: I would not qualify it as "slightly" slower since the final values are still clearly different.
- l.297: lack of parenthesis around Figure.
- l.301: can you confirm it is the case ?

Fig2:

- Please keep the same scales (Fig2A), and correct the typos in the ratio scale in FigB.

Statistical properties of phenotypic selection

- Fig S3: you mention the general model, which is supposed to be the reduced model. However, epsilon is not there.
- Did you select the "best 100" according to the lowest error values ?
- Please mention that it is a log scale.
- After l.315 I think it is Additional Figure (S4).

- I.316 & I.363: "applied the noise" & "apply the model": maybe "applied" is not the right term.

Fig 4:

A) How is the cell growth curve plotted ? It seems like a reunion of segments.

Do you plot the exponential function where the growth parameter was chosen according to the data ?

Can you show this parameter value ?

- In the title: "fit the data to exp growth function" -> it is the other way around, the equation is fitted to the data.

- In the title, you mention "For both experiments", while it is the case only for B.

- I.358: you seem to introduce the cell line already mentioned before.

- I.365: are only the cell death parameters estimated using these data ? In Table 2, all of them are changed.

- I.367 & I.438: please specify that it is the TMZ-mediated cell death parameter.

- I.373: epsilon should have been introduced in the model section. You do not introduce it here.

- I.390: phenotypic.

- I.401: "unusually high variance for early time points": after TMZ administration ? The variance

on this plot does not seem very different. However, is a data point missing at t=200h ?

- Please show the time of TMZ administration on Fig 6.

- I.405 and after: is Fig S5 related to Errors 1&2 only ?

Discussion

- I.431 "regulation" -> expression. "growth, division, &..." (division lacking).

- I.457 "half-inhibition" of what ?

- I.461 & FigS7: when $k_d=0$, there is no TMZ-mediated cell death, therefore the cell viability recovery

has no meaning, and its value put to zero is misleading.

Also, you mention Equations (S1)-(S2) on protein degradations.

- I.471: parameter.

- I.477 to 481: parameters increased/decreased 'with respect to the baseline parameter value'.

- In I.533 you mention the degradation of MGMT alone, while in Fig S2, (S1)-(S2) are mentioned.

- I.548: focused.

Appendix E

Response to Reviewer #1 comments

1- Comment:

- Supplementary material S1, S2. Why is the damaged DNA still transcribed into RNAm and translated into proteins ?

Response

The DNA damage is considered as a global DNA damage term, it is not necessarily the case that the DNA was damaged precisely at the gene encoding for MGMT mRNA. However, by making the stronger assumption that damaged DNA can no longer yield mRNA for MGMT or Reference protein did not impact our results qualitatively and we found that we could recapture our initial results exactly by adjusting the transcription rate.

 I did not understand how relevant it was to keep these equations, since you do not comment on them, and you get rid of them in the reduced model without any discussion. Therefore, since neglecting them does not change qualitatively the results, I would suggest to remove them while justifying their not changing the results in SuppInfo (as for protein degradation). This would allow for more clarity in the model description.

Please either discuss this choice, or simply consider that the damaged DNA can not be transcribed to RNAm in the full model.

Response:

We thank the reviewer for this comment, As mentioned in the paper, we built the first minimalistic, to our knowledge, multiscale stochastic model of MGMT regulation and interaction with TMZ. This model will certainly be expanded by our/different labs, thus maintaining DNA_dam transcription is important to maintain opportunities for investigating the DNA_dam contribution to TMZ induced drug resistance in GBM as the experimental methods for quantifying DNA_dam and the damage location are becoming available. We have added a sentence to the discussion acknowledging this on line line 562.

2- Comment:

- In the steady state approximation, you also assume (it should be explicit) that the repairing action of MGMT is very fast compared with the protein expression dynamics. Is it relevant ?

Response

This is an interesting point. The speed of repairing the DNA damage really depends on the amount of DNA damage and depends on the type, age, and environment of the cell of interest. As a first approximation we assumed it was fast which enabled us to reduce the computational cost of simulating our model by reducing the number of reactions simulated. However, we have now modified Figure S7B in which we begin from a parameter set that yields phenotypic selection in the fast limit and explored the effect of slowing down the DNA repair rate (s) in the full model.

We found that the full model deviated only from the steady state model as the DNA repair rate became very small. We added some sentences about this point on line 481.

Interestingly, we also found that our parameter sets which

produced phenotypic selection required very large values for DNA repair rate (s).

 Thank you for this interesting response. Can you therefore add a sentence when presenting the steady state approximation about this fact (the repairing action assumed also fast, and its relevance)?

Response:

We have now added some of our previous discussion in reply to the reviewer about the steady state on line 191-192.

3- Comment:

- About Eq (S16), now (22): when TMZ acts on DNA, it produces more DNA_Dam, so both signs should be inverted if I am not mistaken.

- *I did not understand Error2 because it was not clear whether the population of cell has constant size in the simulations. How do you compute the fraction of living cells ? Does it mean that dead cells are counted along the simulation ?*

Response

We thank the reviewer for this comment - we have now written some additional sentences explaining that we simulate a cell population of maximum size N on line 154. However, if there are cell death events then this can

drop below N (while subsequent cell division can then replenish the population). The fraction of living cells at time t is computed as follow: $N(t) = M(t)/K$ where $M(t)$ is the number of living cells and K is the total cell population simulated. We have inserted this clarification before introducing Error2 on line 161.

 Therefore, when cell divisions occur, do new cells most frequently replace dead cells ? If the ref [38] given indeed deals with exponentially growing populations, I understood that their point is to compare the analysis of an isolated lineage with the analysis of a snapshot of the population at a given time. I did not understand that they randomly eliminate some cells. Please precise your point regarding this issue, or remove the end of sentence l.155 about better accuracy.

Is it following this constant population assumption that you do not consider a basal cell death in the first "full" model ? Is it so that N is not a bit damped by basal cell death ?

I think this is interesting to the readers, and further indication about the interaction between cell division and cell deaths in the counting could be added.

Response:

We thank the reviewer for this comment and the interest in our modelling strategy. We implemented two different strategies when cell division occurred in the presence of dead cells. One where the dead cells were preferentially replaced by new daughter cells and another where there was no bias and daughter cells randomly replaced any cell in the population. We did not find any significantly different results implementing either strategy so chose the latter for simplicity. We added a couple of sentences beginning on line 155 to reflect this. As we clarify below, we consider a background basal cell death rate in order to capture the cell viability data which decreases even prior to the introduction of TMZ. We also clarified the end of sentence 155 and added an additional reference (cite Ciechonska, M., Sturrock, M., Grob, A., Larrouy-Maumus, G., Shahrezaei, V., & Isalan, M. (2019). Ohm's Law for emergent gene expression under fitness pressure. bioRxiv, 693234.) in order to further clarify our modelling strategy on line 163.

4- Comment:

- Fig 4: why is the curve in panel A not smoothed ? What is this curve and how did you obtain it ? At what times are displayed the histograms?

Response

The curve in Figure 4A refers to our in house experimental growth data. Since we only had data for 7 time points, it appears somewhat disjointed. We have now updated the histograms in Figure 5 to include the precise times as opposed to just "Before TMZ" and "After TMZ".

 I still did not understand how the curve is obtained.

The legend states that it follows from fitting an exponential curve, therefore why not plotting the fitted exponential curve ?

Response:

We thank the reviewer for this comment and we are in agreement here. The plotted curve is the fitted exponential curve in Figure 4.

5- Comment:

- Degradation of MGMT and suicide role: did you estimate again parameters?

Response

Due to limited computational resources we did not redo these estimates. However, we did check the impact of including suicide enzymatic activity and active protein degradation using individual parameter sets from our final posterior distribution estimates and we found the same qualitative behaviour. This is something we will investigate further in future work.

 In my opinion this should be precised in the paper.

Response:

We thank the reviewer for this comment, this is now mentioned in the discussion on line 553.

6- Comment:

- Does the algorithm provide only a univariate analysis, or does it select a vector of parameters ?

Response

A vector of parameters is explored - all parameters are randomly sampled simultaneously. A beneficial aspect of ABC is that we can perform a global sensitivity analysis while simultaneously fitting the model to data.

 Therefore parameters are selected as a vector.

Is the "best fit" defined by the recollection of the modes of each distribution (therefore not taking into account that the combinations of values are selected)? If so, do you observe some vectors θ^* that are close to this choice ?

Response:

We thank the reviewer for this comment. We are using standard ABC parameter inference procedures as in Stumpf et al. (2008) and refer the reader to this paper for further details about this. We note that we found that either taking the modes of the final posterior or one of the final vectors θ^ gave the same results.*

OTHER ISSUES

A) About the cell volume growth

a) In the first paragraph of Background, please explain how you relate growth curves for the numbers of cells with a model for individual cell volume growth.

As I understand it, you fit an exponential model on the data for counts of cells.

Then you use this as a justification to model the volume growth of a single-cell with an exponential law. I do not understand how this is relevant.

Why not simply use a mean time before division to simulate a stochastic cell division process. Then model the effect of TMZ on this quantity.

For cell partitioning, using the same random sampling, one can simulate random partitioning with random bias to produce similar effects.

A typical time before division could perhaps be more realistically related to the cell culture data.

If not, please provide more explanations.

Response:

We thank the reviewer for this suggestion, like all biological processes there are many different approaches that can be taken to model them, we may try the suggested approach in future work. However, we follow the published method of Bertaux et al. and have added this reference to further clarify our method.

b) l.136-137: please explain and discuss how you use the volumes of in vivo tumors over 2 weeks in relation with an equation for single-cell volume growth. Why do you need them ? Is it related to V_I and V_F ? The text you added in the paragraph between (5) and (6) shows that you actually consider that a single-cell has a volume of about 30mL.

Why not instead consider a realistic typical volume of a cell, and assume that it divides when its volume reaches twice the initial value ?

Response:

We thank the reviewer for this comment – we strived to use a value that was relevant to glioblastoma cells. Unfortunately, we accidentally took the wrong value for a single cell. We have now changed this volume from 30 mL to $30 \times 10^{-13} L$. We based this on experimental papers by Watkins et al (2011) and Bryan et al (2014). We added these references and removed the previous one. We note this does not change the results presented, but we have updated the growth rates and values in the Tables. Hence we now consider a realistic typical volume of a single cell and assume that it divides when its volume reaches twice the initial value.

c) However, in Algo 1, you consider $V_i(0)=1$, is it related to V_I ?

Response:

We thank the reviewer for spotting this mistake. We have now changed this to $V_i(0)=15 \times 10^{-13} L$.

B) About referring precisely to the models and methods used.

You introduced two models, and (if I understood properly), you follow two parameter estimation procedures, that you also refer to using the word "model".

The term 'model' is also sometimes found alone, e.g in the conclusion, and it is not clear whether you talk of one of the models, or of your global modeling approach.

I found it still difficult to understand what is used, when, and why.

If your responses show your confidence and knowledge about the procedures followed, I unfortunately miss such clarity in the paper. This is a remark of form, but that is in my opinion critical for understanding and reproducibility.

As an example, following information of Algo 2, the "reduced model" is the "general" model, associated with Errors 1&2 and Table 1, while the "full" model is the "data-constrained model", that is related to Errors 1,2&3 and to Table 2.

However, this is the reduced model that is mentioned in Fig 5 related to calibration using experimental data (i.e Error 3).

-l.202 & algo 1: you seem to say that only the reduced model is simulated.

Please be very specific by using each time the same name and refer to the corresponding equations, and type of calibration or Errors used in the calibration. Make it as specific as possible, even in the Discussion part.

Also:

- the TMZ uptake function is a part of the model, and should be referred to.
- All along the draft, you refer to (4) for the cell volume growth, meaning that you do not consider the inhibition of cell growth by TMZ (introduced in (3)). However, later on you show results related to (3) (FigS3), so that we can not understand what you really considered.
- Please structure the discussion around the main ideas that follow from your work.

Response:

We thank the reviewer for raising these points. We have now made the necessary modifications to remove any ambiguities.

C) Analysis of the results

In my opinion, the work done is very interesting, and there is a lot to discuss. However, there is to me a lack for analysis and interpretation of the results, most likely the observations are simply stated.

For example, when discussing parameter correlations and sensitivity, some correlations are simply mentioned without further analysis (l.313), and no link is made with the sensitivity of the parameters involved, which could be interesting.

Response:

We thank the reviewer for this interesting suggestion. We endeavoured to discuss the results and have added some sentences about how the most sensitive parameters on line 320. We also added in some additional discussion about the bimodality of parameter I_50.

REMARKS AND QUESTIONS

- Please introduce Equation (16) for cell death, and discuss why you choose to neglect basal cell death in the full model while you consider it in the reduced model.

More generally, is it useful to use the full model ? I feel that the comparison made in SuppInfo can be enough, which would allow to perform both types of parameter estimation on the same model.

- l.187 (taking $kd_2=1$). It amounts to having a different parameter value in the numerator of the TMZ-mediated cell death rate (instead of kd). It could be interesting to check whether changing only this value could change a lot the dynamics.

Response:

We thank the reviewer for this comment. We introduced a small basal death rate in the reduced model as we noticed that prior to the introduction of TMZ in our cell viability data that there was some background cell death and our “general” model would not be able to capture this. The full model is very expensive to simulate (many more reaction channels and parameters) and therefore would not be practical for parameter inference. We attempted to vary this numerator parameter but did not find any significant differences in the model dynamics.

- When computing Error1 (the ratio), you compute means over the population separately for REF and MGMT. Did you try computing $\langle \text{REF}/\text{MGMT} \rangle$ to have the ratio in each cell ? It would be interesting to see the histogram of such quantity.

Response:

We thank the reviewer for this comment. We may try this alternative formulation in future work. Preliminary checks of this formalism suggested by the reviewer yielded qualitatively similar results but seemed to be more strongly influenced by outliers.

- I am not very comfortable with the frequent use of the term "need" when commenting the parameter relations selected during the parameter estimation. Can we say that some relation is needed for phenotype selection, when the estimation procedure aimed at minimizing an error function carrying other constraints ?

Response:

We thank the reviewer for this comment. We rephrased this on line 313 to the less strong term “may need”.

- I am not sure to agree with your interpretation for μ and I_{50} in Fig S3. When looking at Errors 1&2, why does TMZ have to inhibit cell volume growth ? Later on you state that I_{50} is not sensitive, does it play a role here ? Also, the correlation plot can be split into two distinct parts, which can be discussed. For example, maybe when I_{50} is too large, then TMZ does not have a big effect for low concentration values, so that the MGMT are less likely to be key players. Maybe the fact that we witness some large values of I_{50} (though less frequently than lower values) is related to Error 2 and cell viability.

Response:

We thank the reviewer for this comment. In the manuscript we refer to papers that show that TMZ inhibits cell volume growth. Since we make TMZ non-zero when running simulations, of course TMZ will inhibit cell volume growth. We found I_{50} to be insensitive for minimising our error. The reviewer raises an interesting observation about I_{50} . We have now added a sentence commenting on the bimodality of the I_{50} posterior, implying that for these particles the Error2 term may dominate in terms of trying to maximise the cell recovery on line 320.

Manual variation of the level of mRNA noise:

- If I understood correctly, you vary the noise and skew levels by changing b_1 . Therefore,

do you take the new_b2 as equal to $\text{old}_b(b_1 \cdot b_2) / \text{new}_b1$?

If (as I understood) the noise and skew are modified together while modifying b_1 , both effects should be investigated together, and therefore maybe a 3D plot would be clearer.

- Fig 3: it could be clearer to use points for each values investigated ?

Response:

We thank the reviewer for this comment. As mentioned in the manuscript we manipulated the mean steady state mRNA levels whilst maintaining the mean steady state protein levels constant and thus vary the mRNA noise and skewness. This is done by decreasing parameter b_1 while increasing the parameter b_2 by the same amount. The plots can be visualised in many different ways each with their own strengths and weaknesses. 3D plots suffer from choosing appropriate viewing angles. We believe the way we have presented it is both printer friendly and demonstrates our points in the clearest manner.

- Did you try without any random bias in partitioning ?

This would allow focusing on the noise related to the protein activity alone.

Response:

We thank the reviewer for this comment. In fact, the simulations related to Figure 3 are actually without accounting for random bias in partitioning, this is now mentioned on line 338. We note that the random bias in partitioning does not change the relationship between noise/skewness and bias in any qualitative way.

- Did you try fixing the most sensitive parameters, and retry the first parameter estimation for the others ?

Response:

We thank the reviewer for this comment. This is not necessary for Bayesian based approaches where all parameters are varied simultaneously. This is not a local method where only one parameter is varied at a time. What the reviewer suggested may be appropriate for point based estimates.

Results

- 1.283-284: "MGMT returns to the initial amount". It seems only true for Fig2-A).

The selection seems reversible for A, and dose-dep and partially reversible for B. These lines could be detailed a bit more.

Response:

We thank the reviewer for this comment. In general we found that while TMZ is continuously 'ON', as in Fig2(B), it was not possible to observe 'complete' reversibility. However, we did find that this 'complete reversibility' is possibly following a short pulse of TMZ. We have now added a sentence to increase the detail of our discussion of this on line 285.

- I.284 It seems like it is the numerical cell doubling time that matters here, not the one for glioma. Can you show your estimate for this value ? It would also support your comment on line 300. Maybe you could define it and explain how you compute it.

Response:

We thank the reviewer for this comment. The cell doubling time can be computed from the growth rate that we estimated using experimental data.

Discussion

- I.434 "All components of the model are necessary". This sentence seems a bit too strong in my opinion. What about epsilon ? The TMZ acting on cell volume growth ?

Response:

We have now rephrased this on line 448 to “All components of the model are necessary (except for basal cell death which was important for capturing our cell viability data).”.

- I.526: if there is a negative feedback loop for transcription, it still leads to lower mRNA values, so more noise. It could however lead to less proteins. Can you explain a bit more how it works?

Response:

We thank the reviewer for this comment. We refer the reviewer to Singh et al. for details, though we note that the counter intuitive aspect of negative feedbacks is that they can decrease noise while also decreasing copy number. Our modelling shows that both of these aspects of negative feedbacks would be beneficial in the context of MGMT related drug resistance.

MINOR ISSUES

Background

- I.90: size -> viability.
- I.100: the cell growth rate and THE impact.
- End of background: please add a conclusion to state the outcome of the work.

Response:

We thank the reviewer for these comments. We have now modified the background section according to reviewers' comments. We believe though that to avoid repetition, 'the outcome of the work' is described in the abstract and we prefer using the last paragraph in the background for detailing the approach and steps.

Modelling cell growth and division

- Eq (3): please introduce [TMZ] and I_50. Is it the pool concentration directly ?
- Also, a 't' is missing in Equation (4).

Response:

We thank the reviewer for these comments. We have now modified this section according to reviewers' comments

- Please precise how the division time of a cell is obtained.

Response:

We thank the reviewer for this comment. We either fixed the growth rate which determines the division time of a cell as we detailed in Table 2 and Figure 5 and 6 (data constrained) or kept it as an open parameter in the general model (Figure S3 and Table 1).

- Eq (5): please precise if the parameters a and b are individual or population parameters.

Response:

We thank the reviewer for this comment. We now clarify that 'a' and 'b' take the same value for all the cells on line 144-145 and as shown in Tables 1 and 2.

- "..., we also simulated our final results...": please be more specific about "final results".

Response:

We thank the reviewer for this comment. We have now changed this to 'we also used a bounded noise approach, for our general and data constrained model simulations, where if a final...?'

- After (5): please do not start a sentence with a symbol (η_1).

-l.165: Following your notations, the cell viability recovery is defined with respect to the cell viability, not the number of living cells. Please correct either the definition or the words used.

Please also define/precise how you obtain the "minimal" number following TMZ addition. Over what is it minimal ?

Response:

We thank the reviewer for these comments. We have now modified this section according to the reviewers' comments.

Modelling intracellular processes

- l.173: if you do not take them into account in the main work, please shift (19) and (20) to the SupplInfo for clarity.

Nevertheless I found that the justification in SupplInfo was satisfying, and prevent any doubts on this assumption.

Response:

We thank the reviewer for this comment. In accordance with the other reviewers' comments, we believe that (19) and (20) must remain in the main manuscript in order to give the reader a clear idea about the model, though we also now refer the readers to the SuppMat for more details about these reactions.

- l.184-185: please introduce the notion of damage by alkylation, for clarity with respect to the rest of the paper.

Response:

We thank the reviewer for this comment. We have now added a reference on line 191-192.

- After (20): is the "pseudo steady-state approximation" the same that the one you call later on "steady-state approximation"?
- p.8, please remove the capital letter at the beginning of the first line.
- After (25), "the reaction defined in (16)".
- l.209 to 217: this part seems more adapted to the Parameter Inference part.

Response:

We thank the reviewer for these comments. We have now modified this section according to reviewers' comments.

Modeling TMZ uptake

- Please put Modeling TMZ uptake before the schematic diagram and the Algo 1, and explain the TMZ pool. What is equal to f_3 or f_4 ?
- The notations of f_3 and f_4 should show if they depend on time.

Response:

We thank the reviewer for these comments. The TMZ pool stands for the TMZ inside an individual cell at time t , we have now added this on line 137. We have now updated figure 1 to reflect the different possible uptake functions, we also stressed the dependency of functions f_3 and f_4 on time, and updated algorithm 1 according to the reviewers comment.

- The notation K is already used.
- The notation x is a bit misleading, perhaps t_{in} and t_{out} could be more specific?
- The inequalities defining f_4 and f_5 are not correct wrt the text.

Response:

We thank the reviewer for these comments. We have now modified this section according to reviewers' comments.

Algo 1:

- In 5., how are the "smallest" indexes defined ?
- In 7., in the exponential, is it τ rather than $t + \tau$? Or $V_i(t)$ should be $V_i(0)$? L_F is V_F ?
- In 8., F is t_f ?
- Equation (4) is not an ODE.

Response:

We thank the reviewer for these comments. Smallest is defined as the minimal value (we use the language here of D. Higham "Modeling and simulating chemical reactions"). We have changed L_F to V_F and F to t_f . We changed "ODE" to "ODE solution".

Parameter estimation

- I.237: "mean following TMZ administration". This is unclear, since this quantity is also computed before TMZ administration.
- I.247 to 250: it would be useful at this point to present what you do next, and maybe to clarify the two procedures you follow.

Response:

That is true that the mean quantity can be computed before TMZ administration, though we only use the time points following TMZ administration in the calculation of the error as is implied. We state what we do next clearly already, i.e. use these errors to calibrate the models using ABC.

Algo 2:

1. Notation epsilon already used.
2. Notation $M(x|\theta)$ rather than x/θ .

- Please precise the value for N_p .

Response:

We thank the reviewer for these comments. We have now modified this section according to the reviewers comments.

Statistical measures

- This part may be more useful at the point where you introduce the variation in mRNA noise.

Response:

We were also of the same opinion initially. But as the reviewer said in his previous comment "Presentation of method, observations on simulations, interpretation and biological discussion are all intertwined". Therefore to address this we moved as much of the methods out of this section as we could. We now firmly believe that the statistical measures section

should be part of the methods section, in this way the reader can familiarise himself with the concepts used later in the manuscript.

Table 2:

- Please clarify the range for the growth rate. Referring to Fig 4 where no range is explicit is not enough.

Response:

We thank the reviewer for this comment. We have now modified the range for the growth rate in Table 2, as the growth rate value is the result of a separate fitting using experimental data, we set range to '-'. We do note though that all the ranges of our priors are explicitly displayed on the x-axes of our correlation plots.

- How did you choose the parameters for TMZ uptake here ?

Response:

We thank the reviewer for this comment. We assume that the reviewer is referring to the f_3 function. We fixed parameters "Q" and "h" in order to have a slow uptake as shown in Figure 5(B). For reproducibility, we now include these parameters in the legend and they are also mentioned in Table 2.

Results

- l.277: f_5 is not mentioned, though used in Figure 2, and commented in line 282.

Also, if you comment the simulations using f_4 first, maybe it is better to switch columns in the Fig ?

Response:

We thank the reviewer for this comment. f_5 is now mentioned on line 284.

- l.279 to 281: this seems not useful at this point. It would be clearer either before when introducing the uptake function, or later on when it is used.

Response:

We thank the reviewer for this comment. We believe that it is good to mention this at this point to stress the fact that we are considering different models of uptake.

- l.296: I would not qualify it as "slightly" slower since the final values are still clearly different.

Response:

We have now reworded this to 'slower' on line 303.

- I.297: lack of parenthesis around Figure.

Response:

We have now added the parenthesis.

- I.301: can you confirm it is the case ?

Response:

We thank the reviewer for this comment. We added a counter to our program to track the number of cell division and cell death events. From this we could confirm that the oscillatory behaviour indicated repeated cell division and death events.

Fig2:

- Please keep the same scales (Fig2A), and correct the typos in the ratio scale in FigB.

Response:

We thank the reviewer for this comment. We have now corrected the typos.

Statistical properties of phenotypic selection

- Fig S3: you mention the general model, which is supposed to be the reduced model. However, epsilon is not there.

Response:

We thank the reviewer for this comment. Epsilon is not a parameter of the general model, but only the reduced one (the data constrained one).

- Did you select the "best 100" according to the lowest error values ?

- Please mention that it is a log scale.

- After I.315 I think it is Additional Figure (S4).

Response:

We thank the reviewer for these comments. The best 100 were selected according to the lowest error values. We did mention that is the log scale in the legend of Figure 3. We have now modified this section according to reviewers' comments.

- I.316 & I.363: "applied the noise" & "apply the model": maybe "applied" is not the right term.

Response:

We thank the reviewer for this comment. We have now rephrased this to 'used'.

Fig 4:

A) How is the cell growth curve plotted ? It seems like a reunion of segments.

Do you plot the exponential function where the growth parameter was chosen according to the data ?

Can you show this parameter value ?

Response:

We thank the reviewer for this comment. We previously addressed this comment. The result of the fitting is shown in Table 2.

- In the title: "fit the data to exp growth function" -> it is the other way around, the equation is fitted to the data.

- In the title, you mention "For both experiments", while it is the case only for B.

Response:

We thank the reviewer for these comments. We modified the Figure 4 legend according to the reviewer's comments.

- I.358: you seem to introduce the cell line already mentioned before.

- I.365: are only the cell death parameters estimated using these data ? In Table 2, all of them are changed.

Response:

We thank the reviewer for these comments. We removed the repetition of introducing the cell line again. We introduced epsilon and re-estimated all the parameters using the data plotted in Figure 4 and provided this data as additional material to the manuscript. We presented the results of this in Table 2.

- I.367 & I.438: please specify that it is the TMZ-mediated cell death parameter.

- I.373: epsilon should have been introduced in the model section. You do not introduce it here.

Response:

We thank the reviewer for these comments. We have now specified TMZ-mediated cell death where appropriate. We note that epsilon is introduced in the reaction (27).

- I.390: phenotypic.

- I.401: "unusually high variance for early time points": after TMZ administration ? The variance on this plot does not seem very different. However, is a data point missing at t=200h ?

Response:

We thank the reviewer for these comments. We refer the reviewer to the additional file containing the cell viability assay results, where the variability is more visible.

- Please show the time of TMZ administration on Fig 6.

Response:

We thank the reviewer for this comment. As mentioned in Figure 6 legend, f_3 is used to simulate TMZ uptake.

- I.405 and after: is Fig S5 related to Errors 1&2 only?

Response:

We thank the reviewer for this comment. As mentioned in the Figure S5 description, Figure S5 is related to Error 1, 2 and 3.

Discussion

- I.431 "regulation" -> expression. "growth, division, &..." (division lacking).

- I.457 "half-inhibition" of what ?

Response:

We thank the reviewer for these comments. We have now modified these lines according to the reviewer's comments.

- I.461 & FigS7: when $k_d=0$, there is no TMZ-mediated cell death, therefore the cell viability recovery has no meaning, and its value put to zero is misleading.

Also, you mention Equations (S1)-(S2) on protein degradations.

Response:

We thank the reviewer for this comment. Using the limiting case of $k_d = 0$ is useful to see the general trend. We can clearly see the cell viability recovery and ratio trending downwards as we reach the point where it is exactly 0. We clarified to the reader though that no cell viability recovery is possible when k_d is exactly equal to zero on line 474.

- I.471: parameter.

- I.477 to 481: parameters increased/decreased 'with respect to the baseline parameter value'.

Response:

We thank the reviewer for these comments. We have now modified these lines accordingly.

- In I.533 you mention the degradation of MGMT alone, while in Fig S2, (S1)-(S2) are mentioned.

Response:

We thank the reviewer for this comment. On line 545 we refer to the suicide enzymatic activity of MGMT, described by (S4).

- I.548: focused.

Response:

This is now corrected.

Appendix F

Response to Reviewer #1 comments

We thank the reviewer for all the time, effort and useful comments which we believe have greatly improved our manuscript.

1- Comment:

About the effect of TMZ on the volume growth.

In all the paper, you refer to Eq (4), which corresponds to a TMZ-free volume growth.

However, in Algo 1, step 7, you explicitly use Eq (1)-(3).

I believe you use Eq (4) only in the absence of TMZ, and use a time discretization of (1)-(3) as soon as there is TMZ. Please clarify this point to the reader, and refer to both types of Equations (not only Eq (4)).

Response:

We have now clarified this point by referring to both types of equations, not only equation (4).

2- Comment:

Eq (22): as already noted previously, the signs should be inverted. Written as it is, in the absence of TMZ the concentration in damaged DNA increases exponentially over time.

Response:

We have now corrected the formulation of Eq (22).

3- Comment:

Please define TMZ in Eqs (22) to (26). It is later defined as the “used TMZ concentration”, which is not clear with respect to [TMZ] (the concentration inside the cell).

Response:

We have now removed ambiguity regarding the TMZ definition in Eq (22)-(26) and reaction (8).

4- Comment:

Algo 2: you state that you use the reduced model (10),(12)-(14),(17),(18), (27) with the general error (1 and 2), while you use the full model (8)-(18) with the data-constraining error (1, 2 and 3).

In your previous answer about Epsilon, you stated “Epsilon is not a parameter of the general model, but only the reduced one (the data constrained one)”.

Table 1 shows that Epsilon is not involved in the reduced model. It is involved in the data-constrained model, which is the full model according to Algo 2.

All further figures and supp figures refer to the same Equations (the reduced model), even if the models used are not the same. This is confusing to the reader. Please clarify in the paper

which model you use in each case. Please also refer to the uptake function you consider in each case, since it also changes.

Response:

We have now corrected the unintentional error in Algo 2 and clarified the definition of the general model on line 194. We removed ambiguities regarding the model definitions and the use of TMZ uptake functions. We also updated Figure 1 to reflect reaction (27).

5- Comment:

Please harmonize notations between μ (supp fig) and μ_0 .

Response:

This is now harmonized, we changed μ to μ_0 in the supp fig S3.

6- Comment:

Line 303: I believe the recovery is not only slower but also smaller.

Response:

This is now mentioned on line 303.

7- Comment:

Line 313: is it the TMZ-mediated cell death rate (kd) ?

Response:

This is now reflected on line 312.

8- Comment:

Line 485: parameter (typo).

Response:

This is now corrected.

9- Comment:

L.515: “general” model ?

Response:

This is now mentioned on line 513.